# Optimal Online Change Detection via Random Fourier Features

**Florian Kalinke**[*]
Information Systems
Karlsruhe Institute of Technology (KIT)
Karlsruhe, Germany
florian.kalinke@kit.edu

**Shakeel Gavioli-Akilagun**[*]
Department of Decision Analytics and Operations
City University Hong Kong
Hong Kong, China
sgavioli@cityu.edu.hk

## Abstract

This article studies the problem of online non-parametric change point detection in multivariate data streams. We approach the problem through the lens of kernel-based two-sample testing and introduce a sequential testing procedure based on random Fourier features, running with logarithmic time complexity per observation and with overall logarithmic space complexity. The algorithm has two advantages compared to the state of the art. First, our approach is genuinely online, and no access to training data known to be from the pre-change distribution is necessary. Second, the algorithm does not require the user to specify a window parameter over which local tests are to be calculated. We prove strong theoretical guarantees on the algorithm's performance, including information-theoretic bounds demonstrating that the detection delay is optimal in the minimax sense. Numerical studies on real and synthetic data show that our algorithm is competitive with respect to the state of the art.

## 1 Introduction

In the online change point detection problem, data is observed sequentially, and the goal is to flag a change if the distribution of the data changes. The problem dates back to the early work of Page [43], and has now been extensively studied in the statistics and machine learning literature [30, 3, 52]. However, these classical approaches assume that the data are low-dimensional and that the pre- and post-change distributions belong to a known parametric family. In modern applications, both assumptions are usually not satisfied. Examples of modern online change point detection problems include: detecting changes in audio streams [4], in videos [1, 25], in highway traffic data [15], in internet traffic data [31], or in cardiac time series [61]. Further, such data frequently has high volume, and online change point detection procedures should still be able to process new data in real time and with limited memory.

While algorithms for the problem of non-parametric online change point detection have been proposed—see Section 2 for a brief overview—state-of-the-art methods suitable for modern data suffer from at least one of the following two limitations. First, most procedures are not genuinely online from a statistical perspective, in the sense that they either assume the pre-change distribution to be known completely or they assume having access to historical data known to be from the pre-change distribution. Second, most approaches require the user to specify a window parameter over which local tests for a change in distribution will be applied. Choosing such a window is notoriously difficult [47], and choosing the window too small or too large leads to a reduction in power or an increase in the detection delay, respectively. Moreover, the window size imposes a limit on the historic data considered, rendering detecting sufficiently small changes impossible.

---

[*]Contributed equally.

39th Conference on Neural Information Processing Systems (NeurIPS 2025).

Motivated by the above limitations and challenges, we propose a new algorithm called Online RFF-MMD (Random Fourier Feature Maximum Mean Discrepancy). On a high level, the algorithm performs sequential two-sample tests based on the kernel-based maximum mean discrepancy (MMD; [53, 14]). Crucially, approximating the maximum mean discrepancy using random Fourier features (RFFs; [45]) leads to a detection statistic that can be computed in linear and updated in constant time. By embedding these local tests in a sequential testing scheme on a dyadic grid of candidate change point locations, we obtain an algorithm that does not require a window parameter and features a time and space complexity logarithmic in the amount of data observed.

This article makes the following contributions:

- **Computational efficiency:** We propose Online RFF-MMD, a fully non-parametric change point detection algorithm that does not require access to historical data, has no window parameter, and features logarithmic runtime and space complexity.
- **Minimax optimality:** Online RFF-MMD comes with strong theoretical guarantees. In particular, we derive information-theoretic bounds showing that the detection delay incurred by Online RFF-MMD is optimal up to logarithmic terms in the minimax sense. While related results are known in the offline setting [41, 42] and in the parametric online setting [30, 64], ours is the first result of this kind for kernel-based online change point detection.
- **Empirical validation:** We perform a suite of benchmarks on synthetic data, the MNIST data, and the HASC data to demonstrate the applicability of the proposed method. Our approach achieves competitive results throughout all experiments.

The article is structured as follows. We recall related work in Section 2 and introduce our notations and the problem in Section 3. Section 4 presents our algorithm and its guarantees, and Section 5 its minimax optimality. Experiments are in Section 6 and limitations in Section 7. Additional results and all proofs are in the appendices.

Table 1: Comparison of kernel-based change detectors. Genuinely online — whether the algorithm can be executed without reference data known to be from the pre-change distribution; Window free — whether the algorithm requires selection of a window parameter over which the detection statistic is calculated; Time comp. — runtime complexity per new observation; Space comp. — total space complexity; $n$ — total number of observations.[2]

| Algorithm | Genuinely online | Window free | Time comp. | Space comp. |
|---|---|---|---|---|
| Scan $B$-statistics [33] | ✗ | ✗ | $\mathcal{O}(NW^2)$ | $\mathcal{O}(NW)$ |
| NEWMA [24] | ✓ | ✗ | $\mathcal{O}(r)$ | $\mathcal{O}(r)$ |
| Online kernel CUSUM [59] | ✗ | ✗ | $\mathcal{O}(NW^2)$ | $\mathcal{O}(NW)$ |
| **Online RFF-MMD** | ✓ | ✓ | $\mathcal{O}(r\log n)$ | $\mathcal{O}(r\log n)$ |

## 2 Related work

In the case of univariate data, numerous methods for non-parametric online change point detection have been proposed; we refer to Ross [49] for a more comprehensive overview. The most successful among these approaches exploit the fact that all information is contained in the data's empirical distribution function, which is a functional of the ranks. Rank-based online change point detection methods have been proposed by [13, 19, 48]. However, their extension to multivariate data is challenging as this requires a computationally efficient multivariate analogue to scalar ranks.

To tackle change point detection on multivariate data, many approaches exist; see Wang and Xie [58] for a survey. For example, Yilmaz [62], Kurt et al. [28] introduce procedures using summary statistics based on geometric entropy minimization, but assume knowledge of the pre-change distribution. An alternative non-parametric approach is using information on distances between data points [7, 8]. However, here one can construct alternatives which the procedures will always fail to detect as the limits of the test statistics employed do not metrize the space of probability distributions.

---

[2]In Table 1, where applicable, $r$ denotes the number of random Fourier features, $W$ denotes the size of the window, and $N$ denotes the number of blocks of historical data. We treat the dimension of the data as fixed.

One principled approach to tackle this challenging setting is using kernel-based two sample tests via the MMD, which we recall briefly in Section 3.2. The key property, if the underlying kernel is characteristic [11, 55], is that the MMD metrizes the space of probability distributions and thus allows detecting any change. However, the sample estimators of the MMD typically have a quadratic runtime complexity, which prohibits their direct application in the online setting. To overcome this challenge, Zaremba et al. [66] propose Scan $B$-statistics, which achieve sub-quadratic time complexity by splitting the data into blocks and computing the quadratic-time MMD on each block. Consequently, Li et al. [32, 33] introduce an online change point detection algorithm that recursively estimates the Scan $B$-statistic over a sliding window. Extending this idea, Wei and Xie [59] propose an algorithm which performs the same operation over grid of windows of different sizes. However, these algorithms all require the choice of a window parameter and are not genuinely online as they require historical data known to be from the pre-change distribution. Keriven et al. [24] propose comparing two exponentially smoothed MMD statistics, where the MMD is approximated by using random Fourier features. However, the window selection problem is not avoided because, as shown by the authors, the smoothed statistic can be interpreted as computing differences between MMDs calculated on two windows of different sizes.

We summarize the kernel-based approaches coming with theoretical guarantees in Table 1 and note that we present two extensions of our proposed Online RFF-MMD approach in the appendix. The first allows taking historical data into account (Appendix A.1). The second permits detecting multiple change points within the same stream, with theoretical guarantees (Appendix A.2).

## 3 Preliminaries

In this section, we formally introduce the online change point detection problem (Section 3.1) and recall kernel-based two sample testing together with its RFF-based approximation (Section 3.2).

### 3.1 Problem statement

We consider a data stream $X_1, X_2, \ldots$ observed online, where the $X_t$-s are independent random variables taking values in $\mathbb{R}^d$, with $d$ arbitrary but fixed. Let $\mathbb{P}, \mathbb{Q} \in \mathcal{M}_1^+ := \mathcal{M}_1^+\big(\mathbb{R}^d\big)$ and $\mathbb{P} \neq \mathbb{Q}$, where $\mathcal{M}_1^+\big(\mathbb{R}^d\big)$ denotes the set of all Borel probability measures on $\mathbb{R}^d$. We assume that there exists an $\eta \in \mathbb{N} := \{1, 2, \ldots\}$ such that

$$X_t \sim \begin{cases} \mathbb{P} & \text{for } t = 1, \ldots, \eta \\ \mathbb{Q} & \text{for } t = \eta + 1, \eta + 2, \ldots \end{cases}.$$

The goal is to stop the process with minimal delay as soon as $\eta$ is reached, but not before. Note that we may have $\eta = \infty$ in which case the process should never be stopped. Formally, one wants to test

$H_{0,n} : X_t \sim \mathbb{P}$ for each $t \leq n$ and some $\mathbb{P} \in \mathcal{M}_1^+$ versus

$H_{1,n} : \exists \eta < n$ s.t. $X_t \sim \begin{cases} \mathbb{P} & \text{if } 1 \leq t \leq \eta \\ \mathbb{Q} & \text{if } \eta < t \leq n \end{cases}$, and some $\mathbb{P}, \mathbb{Q} \in \mathcal{M}_1^+$ where $\mathbb{P} \neq \mathbb{Q}$,

for each $n \in \mathbb{N}$, until a local null is rejected. A secondary aim, once a local null has been rejected, is to accurately estimate $\eta$. Solving the aforementioned problems boils down to constructing an extended stopping time $N$, which, in a sense we make precise in Section 4.3, is close to $\eta$. Let $\mathcal{F}_t = \sigma(X_s \mid s = 1, \ldots, t)$ be the natural filtration generated by the $X_s$'s up to time $t$ and recall that a random variable $N$ is an extended stopping time if (i) $N$ takes vales in $\mathbb{N} \cup \{\infty\}$ and (ii) for each $t \in \mathbb{N}$ the event $\{N \leq t\}$ is $\mathcal{F}_t$-measurable.

Minimizing the distance between $\eta$ and $N$ is analogous to maximizing the power of a particular sequential testing procedure, and it is therefore natural to impose some conditions on the sequential testing analogue of statistical size; we recall the two most frequent ones in the following. In the sequel, let $\mathbb{P}_k$ be the joint distribution of $\{X_t\}_{t>0}$ when $\{\eta = k\}$ ($k \in \mathbb{N}$), and let $\mathbb{E}_k$ be the expectation under this distribution.

1. The average run length until a spurious rejection under the global null should be bounded from below by a chosen quantity [36]. Specifically, for a given $\gamma > 1$ it should hold that

$$\mathbb{E}_\infty[N] \geq \gamma. \tag{1}$$

2. The uniform false alarm probability should be bounded from above by a chosen quantity [30]. Specifically, for a given $\alpha \in (0, 1)$ it should hold that

$$\mathbb{P}_\infty(N < \infty) \leq \alpha. \tag{2}$$

We show in Section 4.3 that Online RFF-MMD is able to satisfy either of the conditions (1) or (2).

### 3.2 Fast kernel-based two sample tests

To resolve the change point detection problem (Section 3.1), we embed fast two sample tests based on RFF approximations to the MMD into a particular sequential testing scheme. In this section, we briefly recall the MMD statistic and its RFF approximation.

Let $\mathcal{H}_K$ be a reproducing kernel Hilbert space (RKHS; [2, 56]) on $\mathbb{R}^d$ with (reproducing) kernel $K : \mathbb{R}^d \times \mathbb{R}^d \to \mathbb{R}$. Denote by $\text{supp}(\Lambda) = \overline{\{A \in \sigma(\mathbb{R}^d) \mid \Lambda(A) > 0\}}$ the support of a Borel measure $\Lambda$ on $\mathbb{R}^d$, where $\overline{A}$ denotes the closure of a set $A$ and $\sigma(\mathbb{R}^d)$ the Borel $\sigma$-algebra on $\mathbb{R}^d$. Throughout the article, we make the following assumption on the kernel:

**Assumption 1.** *The kernel $K : \mathbb{R}^d \times \mathbb{R}^d \to \mathbb{R}$ is non-negative, continuous, bounded ($\exists B > 0$ s.t. $\sup_{\mathbf{x} \in \mathbb{R}^d} K(\mathbf{x}, \mathbf{x}) \leq B$), translation-invariant ($K(\mathbf{x}, \mathbf{y}) = \psi(\mathbf{x} - \mathbf{y})$ for some positive definite $\psi : \mathbb{R}^d \to \mathbb{R}$), and characteristic ($\text{supp}(\Lambda) = \mathbb{R}^d$ with $\psi(\mathbf{x}) = \int e^{-i\omega^\mathsf{T}\mathbf{x}} \mathrm{d}\Lambda(\omega)$).*

To simplify exposition, we also assume that $K(\mathbf{0}, \mathbf{0}) = 1$, which can be achieved by scaling any bounded kernel. The conditions in Assumption 1 are satisfied by several commonly used kernels, including the Gaussian kernel, mixtures of Gaussians, inverse multi-quadratic kernels, Matérn kernels, Laplace kernels, or B-spline kernels [55]. Assumption 1 permits to approximate the respective kernel function by finite-dimensional feature maps through Bochner's theorem (elaborated below), which, in turn, allows the effective online estimation of MMD detailed in Section 4.

To any $\mathbb{P} \in \mathcal{M}_1^+$, one can associate the kernel mean embedding $\mu_K(\mathbb{P}) \in \mathcal{H}_K$, taking the form[3]

$$\mu_K(\mathbb{P}) = \int_{\mathbb{R}^d} K(\cdot, \mathbf{x}) \mathrm{d}\mathbb{P}(\mathbf{x}), \tag{3}$$

where the integral is meant in Bochner's sense [10, Chapter II.2]. The continuity and boundedness assumptions ensure the existence of $\mu_K(\mathbb{P})$ for any $\mathbb{P} \in \mathcal{M}_1^+$ [55, Proposition 2]. Expression (3) gives rise to the MMD, which quantifies the distance between two measures $\mathbb{P}, \mathbb{Q} \in \mathcal{M}_1^+$ via the distance between their mean embeddings in terms of the RKHS norm, and takes the form

$$\text{MMD}_K[\mathbb{P}, \mathbb{Q}] = \|\mu_K(\mathbb{P}) - \mu_K(\mathbb{Q})\|_{\mathcal{H}_K}. \tag{4}$$

Crucially, the characteristic assumption ensures that (3) is injective and that the MMD metrizes the space $\mathcal{M}_1^+$ [55, Theorem 9], implying $\text{MMD}_K[\mathbb{P}, \mathbb{Q}] = 0$ iff. $\mathbb{P} = \mathbb{Q}$. Given samples $X_1, \ldots, X_n \overset{\text{i.i.d.}}{\sim} \mathbb{P}$ and $Y_1, \ldots, Y_m \overset{\text{i.i.d.}}{\sim} \mathbb{Q}$ with associated empirical measures $\hat{\mathbb{P}}_n =: X_{1:n}$ and $\hat{\mathbb{Q}}_m =: Y_{1:m}$, respectively, the squared plug-in estimator of $\text{MMD}_K[\mathbb{P}, \mathbb{Q}]$ takes the form

$$(\text{MMD}_K[X_{1:n}, Y_{1:m}])^2 = \left\|\mu_K\left(\hat{\mathbb{P}}_n\right) - \mu_K\left(\hat{\mathbb{Q}}_n\right)\right\|_{\mathcal{H}_K}^2 = \left\|\frac{1}{n}\sum_{i=1}^n K(\cdot, X_i) - \frac{1}{m}\sum_{i=1}^m K(\cdot, Y_i)\right\|_{\mathcal{H}_K}^2$$

$$= \frac{1}{n^2}\sum_{i=1}^n \sum_{j=1}^n K(X_i, X_j) + \frac{1}{m^2}\sum_{i=1}^m \sum_{j=1}^m K(Y_i, Y_j) - \frac{2}{nm}\sum_{i=1}^n \sum_{j=1}^m K(X_i, Y_j). \tag{5}$$

The computation of (5) costs $\mathcal{O}\left((\max(m, n))^2\right)$, rendering its use in an online testing procedure computationally infeasible.

Random Fourier features [45, 54] alleviate this bottleneck in the offline setting; we recall the method in the following. For some $\omega \in \mathbb{R}^d$ write $\zeta_\omega(\mathbf{x}) = e^{-i\omega^\mathsf{T}\mathbf{x}}$, where $i = \sqrt{-1}$. By Bochner's theorem,

$$K(\mathbf{x}, \mathbf{y}) = \psi(\mathbf{x} - \mathbf{y}) = \int_{\mathbb{R}^d} e^{-i\omega^\mathsf{T}(\mathbf{x}-\mathbf{y})} \mathrm{d}\Lambda(\omega) = \mathbb{E}_{\omega \sim \Lambda}\left[\zeta_\omega(\mathbf{x})\zeta_\omega(\mathbf{y})^*\right],$$

---

[3]For $\mathbf{x} \in \mathbb{R}^d$, $K(\cdot, \mathbf{x}) : \mathbb{R}^d \to \mathbb{R}$ denotes the map $\mathbf{x}' \mapsto K(\mathbf{x}', \mathbf{x})$.

where $*$ denotes the complex conjugate and, using that $K(0,0) = 1$, one has that $\Lambda \in \mathcal{M}_1^+$. As $\Lambda$ and $K$ are real-valued, Euler's identity implies that $\int e^{-i\omega^\mathsf{T}(\mathbf{x}-\mathbf{y})} \mathrm{d}\Lambda(\omega) = \int \cos\left(\omega^\mathsf{T}(\mathbf{x}-\mathbf{y})\right) \mathrm{d}\Lambda(\omega)$. Therefore, using that $\cos(\alpha - \beta) = \cos\alpha\cos\beta + \sin\alpha\sin\beta$, picking some $r \in \mathbb{N}$, and sampling $\omega_1, \ldots, \omega_r \overset{\text{i.i.d.}}{\sim} \Lambda$, a low variance estimator for $K(\mathbf{x}, \mathbf{y})$ is given by

$$\hat{K}(\mathbf{x}, \mathbf{y}) := \langle \hat{z}_K(\mathbf{x}), \hat{z}_K(\mathbf{y}) \rangle, \text{ where } \hat{z}_K(\mathbf{x}) = \frac{1}{\sqrt{r}} \left( (\sin(\omega_j^\mathsf{T}\mathbf{x}), \cos(\omega_j^\mathsf{T}\mathbf{x})) \right)_{j=1}^r \in \mathbb{R}^{2r}. \quad (6)$$

By noting that $\hat{K} : \mathbb{R}^d \times \mathbb{R}^d \to \mathbb{R}$ is the kernel associated with the RKHS $\mathcal{H}_{\hat{K}} = \mathbb{R}^{2r}$, we may approximate (5) by

$$\mathrm{MMD}_{\hat{K}}[X_{1:n}, Y_{1:m}] = \left\| \mu_{\hat{K}}\left(\hat{\mathbb{P}}_n\right) - \mu_{\hat{K}}\left(\hat{\mathbb{Q}}_n\right) \right\|_{\mathcal{H}_{\hat{K}}} = \left\| \frac{1}{n} \sum_{i=1}^n \hat{z}_K(X_i) - \frac{1}{m} \sum_{i=1}^m \hat{z}_K(Y_i) \right\|_2. \quad (7)$$

Importantly, as the mean embeddings in (7) are Euclidean vectors, their distance can be computed with the standard Euclidean norm. This leads to a statistic which can computed in linear time and updated in constant time, allowing its use for sequential testing.

## 4 Online change point detection via random Fourier features

In this section, we present our proposed stopping time for resolving the change point detection problem. In particular, we give a precise definition in Section 4.1, an efficient algorithm in Section 4.2, and theoretical guarantees in Section 4.3.

### 4.1 The RFF-MMD stopping time

The intuitive construction of our RFF-MMD stopping time is as follows. We begin by choosing an $r \in \mathbb{N}$ and a kernel $K$, and construct its RFF approximation (6) using $r$ random features. For every $n \geq 2$, having observed data $\{X_1, \ldots, X_n\}$, we consider $\log_2 n$ possible sample splits of the domain $\{1, \ldots, n\}$ at locations $n - 2^j$ with $j = 0, \ldots, \lfloor \log_2 n \rfloor - 1$. For every such split, we approximate the MMD between the two samples using (7). A change is declared at the first $n$ for which at least one such statistic, appropriately normalized so that it is $\mathcal{O}_P(1)$ under its local null, is larger than a given threshold. Formally, we have

$$N = \inf \left\{ n \geq 2 \mid \bigcup_{j=0}^{\lfloor \log_2(n) \rfloor - 1} \sqrt{\frac{2^j(n - 2^j)}{n}} \mathrm{MMD}_{\hat{K}}\left[X_{1:(n-2^j)}, X_{(n-2^j+1):n}\right] > \lambda_n \right\}, \quad (8)$$

where $\{\lambda_n \mid n \in \mathbb{N}\}$ is a non-decreasing sequence that we make precise in Section 4.3, taking requirements (1) or (2) into account.

The first use of an exponential grid in online change point detection appears to be due to Lai [29]. Recently, similar techniques have been used by Yu [63], Kalinke et al. [23], Moen [39]. The dyadic grid used in (8) has two advantages. First, only a logarithmic number of tests must be performed with each new observation. Second, the grid is sufficiently dense, so the obtained stopping time has essentially the same behavior as the computationally infeasible variant, which considers every possible candidate change point location.

### 4.2 The RFF-MMD algorithm

We now present an efficient implementation of (8) and analyze its runtime and space complexity. We show the pseudo code of our proposed method in Algorithm 1; see also Example 1 and Figure 1 for a summary. The details are as follows. For each new observation $X_t$, we create a new window $W$, storing $z = \hat{z}_K(X_t)$ and $c = 1$ (Lines 3–4). The window $W$ is then added to the list of all windows $\mathcal{W}$ (Line 5). The remaining algorithm has two main parts.

1. **Change point detection.** To detect changes, we iterate all $|\mathcal{W}| - 1$ dyadic points $i$ (Line 6), and, for each $i$, merge the feature maps coming before $i$ and coming after $i$ (along with their counts) to compute the MMD statistic (Lines 7–9). If the statistic exceeds the threshold (Line 10; see Section 4.3 for its value), a change is flagged and we drop the data coming before the change.

2. **Structure maintenance.** To set up and maintain the dyadic structure, we merge windows that have the same counts (Line 16) by first summing their $z$-s and their $c$-s (Lines 17–18), and then replacing them in the list of windows $\mathcal{W}$ accordingly (Line 19). We note that pop removed the windows beforehand.

---

**Algorithm 1** Online RFF-MMD change point detection

---

**Input:** Stream $X_1, X_2, \ldots$ and a sequence of thresholds $\{\lambda_t \mid t \in \mathbb{N}\}$.
**Output:** Changepoint location and detection time.
  1: $\mathcal{W} \leftarrow$ empty list
  2: **for** $X_t \in X_1, X_2, \ldots$ **do**                                                          ▷ Main loop
  3:      $W.z \leftarrow \hat{z}_K(X_t)$
  4:      $W.c \leftarrow 1$
  5:      $\mathcal{W} \leftarrow \mathcal{W}.\text{append}(W)$
  6:      **for** $i \in 1, \ldots, |\mathcal{W}| - 1$ **do**                                   ▷ Detect changes
  7:           $c_1 \leftarrow \sum_{j=i+1}^{|\mathcal{W}|} \mathcal{W}_j.c$
  8:           $c_2 \leftarrow \sum_{j=1}^{i} \mathcal{W}_j.c$
  9:           $\text{MMD}_{\hat{K}} \leftarrow \left\| \frac{1}{c_1} \sum_{j=i+1}^{|\mathcal{W}|} \mathcal{W}_j.z - \frac{1}{c_2} \sum_{j=1}^{i} \mathcal{W}_j.z \right\|_2$
 10:           **if** $\sqrt{\frac{c_1 c_2}{c_1 + c_2}} \, \text{MMD}_{\hat{K}} > \lambda_t$ **then**
 11:               **print** Change detected at element $X_t$; most likely at position $i$.
 12:               **return**
 13:      **while** $|\mathcal{W}| \geq 2$ **do**                                   ▷ Maintain exponential structure
 14:           $W_1 \leftarrow$ pop $\mathcal{W}$
 15:           $W_2 \leftarrow$ pop $\mathcal{W}$
 16:           **if** $W_1.c = W_2.c$ **then**
 17:               $W.c \leftarrow W_1.c + W_2.c$
 18:               $W.z \leftarrow W_1.z + W_2.z$
 19:               $\mathcal{W} \leftarrow \mathcal{W}.\text{append}(W)$
 20:           **else**
 21:               $\mathcal{W} \leftarrow \mathcal{W}.\text{append}(W_1).\text{append}(W_2)$
 22:               **break**

---

We now analyze the runtime and space complexity of Online RFF-MMD. For each insert operation, Algorithm 1 performs three steps, which we analyze independently.

1. **Setup.** The computation of $\hat{z}_K(X_t)$, defined in (6), requires computing $2r$ trigonometric functions of $d$-dimensional inner products and thus is in $\mathcal{O}(rd)$.

2. **Change point detection.** The dominating term is computing $\text{MMD}_{\hat{K}}$, which requires $\mathcal{O}(|\mathcal{W}|r)$ computations. Repeating the computation $|\mathcal{W}|$ times leads to a cost of $\mathcal{O}(|\mathcal{W}|^2 r)$. The calculation of the threshold is in $\mathcal{O}(1)$, which gives an overall cost of $\mathcal{O}(|\mathcal{W}|^2 r)$. However, we note that memoization of all sums allows to implement the change point detection in a single sweep over $\mathcal{W}$ (at each step, the attributes of one $W \in \mathcal{W}$ are subtracted from one sum and added to another sum) and thereby reduces the runtime complexity to $\mathcal{O}(|\mathcal{W}|r)$.

3. **Maintenance.** In the worst case, $\mathcal{O}(|\mathcal{W}|)$ merge operations need to be performed. Each merge requires $\mathcal{O}(r)$ operations, which yields a total cost of $\mathcal{O}(|\mathcal{W}|r)$.

Adding the results obtained in steps 1.–3. shows that the algorithm has an overall runtime complexity of $\mathcal{O}(|\mathcal{W}|r) = \mathcal{O}(r \log n)$ per insert operation. As the algorithm stores, for each $W \in \mathcal{W}$, a number ($W.c$) and a vector ($W.z \in \mathbb{R}^{2r}$), the total space complexity when having observed $n$ samples is $\mathcal{O}(|\mathcal{W}|r) = \mathcal{O}(r \log n)$. We note that $r$ is a fixed parameter in practice and thus constant.

The following Example 1 and the corresponding Figure 1 illustrate how Algorithm 1 operates upon observing the first 6 samples $X_1, \ldots, X_6$.

**Example 1.** *When observing the first element $X_1$, the algorithm creates a new window $W$, storing the feature map $\hat{z}_K(X_1)$ and that $W$ has one element ($c = 1$). Similarly, when observing $X_2$, the algorithm creates a new window $W'$, storing $\hat{z}_K(X_2)$ and $c = 1$. As both windows, $W$ and $W'$, have the same counts, the algorithm merges them into a new window $W$, storing $\hat{z}_K(X_1) + \hat{z}_K(X_2)$ and $c = 2$, thereby maintaining the dyadic structure. The algorithm proceeds in this manner, resulting*

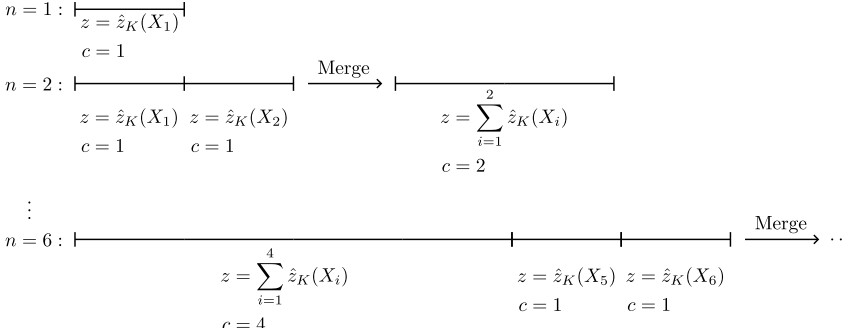

Figure 1: Schematic representation of the proposed algorithm upon observing the first $n = 6$ elements. Merging equal sized "windows" yields the division along dyadic points.

*in the construction of the dyadic grid outlined in Section 4.1. For example, when observing $X_6$, there are again two windows of size $1$, which the algorithm merges to store a total of two windows, one capturing $X_1, \ldots, X_4$ and the other one capturing $X_5, X_6$. We recall that the observations themselves are never stored explicitly.*

Before presenting out theoretical results in the following section, we emphasize that Algorithm 1 matches (8), that is, there is no approximation error.

## 4.3 Theoretical results

In this section, we analyze the theoretical behavior of the RFF-MMD algorithm. We first study the behavior of the stopping time defined in (8) under the global null of no change. Theorems 1 and 2 show that with an appropriately chosen sequence of thresholds the stopping time can be made to attain, respectively, a desired average run length (1) or a desired uniform false alarm probability (2).

**Theorem 1.** *Let $N$ be the extended stopping time defined via (8). For any $\gamma > 1$, if the sequence of thresholds satisfies $\lambda_n \geq \sqrt{2} + \sqrt{2 \log\left(4\gamma \log_2\left(2\gamma\right)\right)}$ for all $n \in \mathbb{N}$, it holds that $\mathbb{E}_\infty[N] \geq \gamma$.*

**Theorem 2.** *Let $N$ be the extended stopping time defined via (8). For any $\alpha \in (0, 1)$, if the sequence of thresholds satisfies $\lambda_n \geq \sqrt{2} + \sqrt{2\left(\log(n/\alpha) + 2\log\left(\log_2(n)\right) + \log\left(\log_2(2n)\right)\right)}$ for each $n \in \mathbb{N}$, it holds that $\mathbb{P}_\infty(N < \infty) \leq \alpha$.*

We emphasize that these guarantees do not depend on the number of random features used in constructing (6) and the bounds on the threshold sequences do not require any knowledge of the pre-change distribution.

Next, we study the detection delay incurred by (8) when the threshold sequence is chosen to control the uniform false alarm probability at some level $\alpha \in (0, 1)$. In the following, we assume that the data take values in a compact subset of $\mathbb{R}^d$, and denote the Lebesgue measure of a set by $|\cdot|$. The following result shows that with high probability, provided the number of RFFs is chosen sufficiently large, the detection delay incurred by (8) is bounded from above by a quantity depending only on the chosen $\alpha$, the number of pre-change observations, and the squared MMD between the pre- and post-change distributions.

**Theorem 3.** *Let $N$ be the extended stopping time defined via (8) with threshold sequence $\{\lambda_n \mid n \in \mathbb{N}\}$ defined as in Theorem 2 for a chosen $\alpha \in (0, 1)$. If $\operatorname{supp}(\mathbb{P}) \cup \operatorname{supp}(\mathbb{Q}) \subseteq \mathcal{X}$ for some compact set $\mathcal{X} \subset \mathbb{R}^d$, the quantities $\eta$, $\alpha$, and $\operatorname{MMD}_K[\mathbb{P}, \mathbb{Q}]$ jointly satisfy*

$$\eta \geq C_1 \frac{\log\left(2\eta/\alpha\right)}{\left(\operatorname{MMD}_K[\mathbb{P}, \mathbb{Q}]\right)^2}, \tag{9}$$

*and the number of random features in (7) is chosen so that*

$$\sqrt{r} \geq C_2 \frac{h(d, |\mathcal{X}|, \sigma) + \sqrt{2\log\left(2/\alpha\right)}}{\left(\operatorname{MMD}_K[\mathbb{P}, \mathbb{Q}]\right)^2}, \tag{10}$$

*then with probability at least $1 - \alpha$, it holds that*

$$(N - \eta)^+ \leq 1 \vee C_3 \frac{\log(2\eta/\alpha)}{(\mathrm{MMD}_K[\mathbb{P}, \mathbb{Q}])^2}, \tag{11}$$

*where $C_1$, $C_2$, and $C_3$ are absolute constants independent of $\eta$, $\alpha$, and $\mathrm{MMD}_K[\mathbb{P}, \mathbb{Q}]$, and, with $\sigma^2 = \int_{\mathbb{R}^d} \|\omega\|_2^2 \, \mathrm{d}\Lambda(\omega)$, we have put*

$$h(d, |\mathcal{X}|, \sigma) = 23\sqrt{2d\log(2|\mathcal{X}|+1)} + 32\sqrt{2d\log(\sigma+1)} + 16\sqrt{2d\left[\log(2|\mathcal{X}|+1)\right]^{-1}},$$

*which is likewise independent of $\eta$, $\alpha$, and $\mathrm{MMD}_K[\mathbb{P}, \mathbb{Q}]$.*

Condition (9) can be interpreted as a signal strength requirement, measuring the strength according to the number of observations from the pre-change distribution and the squared MMD between $\mathbb{P}$ and $\mathbb{Q}$. The term $\log(2\eta/\alpha)$ reflects the cost of multiple testing when the data are drawn from $\mathbb{P}$. Such requirements are unavoidable from the minimax perspective in the corresponding offline problem [63, 42, 41], and the discussion in Yu et al. [64, Section 4.1] suggests that the same is true for genuinely online change point problems.

The requirement on the number of RFFs in (10) depends on $\mathrm{MMD}_K[\mathbb{P}, \mathbb{Q}]$, which is unknown in practice. However, if one assumes an asymptotic setting with a fixed distance between $\mathbb{P}$ and $\mathbb{Q}$, and $\alpha \downarrow 0$, then (10) suggests that the number of RFFs should be chosen as $r = \Theta(\log 1/\alpha)$. To put this result into perspective, we compare it to online change procedures having a window. Here, the optimal window length also depends on the distance between the pre- and post-change distributions [33, 59]. However, choosing the window larger than this quantity can lead to an increase in the detection delay. This is not the case for RFF-MMD: in practice, one may choose the number of RFFs as large as possible subject to computational constraints, and choosing a larger number of RFFs does not negatively impact the detection delay.

## 5   Minimax optimality of RFF-MMD

Recall that with the conditions of Assumption 1, the underlying kernel is characteristic and $\mathrm{MMD}_K$ metrizes the space $\mathcal{M}_1^+$. Therefore, $\mathrm{MMD}_K[\mathbb{P}, \mathbb{Q}] > 0$ for any $\mathbb{P} \neq \mathbb{Q}$ and we have that (11) guarantees that our stopping time (8) obtains a finite detection delay, with high probability for any fixed alternative. Still, one may ask whether the detection delay is optimal. The following theorem resolves this question and shows that the detection delay of RFF-MMD is essentially optimal from a minimax perspective, up to logarithmic terms.

**Theorem 4.** *For every kernel $K : \mathbb{R}^d \times \mathbb{R}^d \to \mathbb{R}$ satisfying Assumption 1 there is a constant $C_K$ depending only on $K$ and absolute constants $\alpha_0, \beta_0 \in (0,1)$ independent of $K$, such that for any $\alpha \leq \alpha_0$ it holds that*

$$\inf_{N : \mathbb{P}_\infty(N<\infty) \leq \alpha} \sup_{\substack{\eta > 1 \\ \mathbb{P}, \mathbb{Q} \in \mathcal{M}_1^+}} \mathbb{P}_\eta \left( N \geq \eta + C_K \frac{\log(1/\alpha)}{(\mathrm{MMD}_K[\mathbb{P}, \mathbb{Q}])^2} \right) \geq \beta_0 \tag{12}$$

*with the infimum being over all extended stopping times.*

We remark that in the online change point detection literature [40, 46], it is more common to study the expected risk of a stopping time. For example, for fixed $\mathbb{P}, \mathbb{Q}$, it is common to work with the so-called worst-worst-case average detection delay [37] of a given stopping time $N$, which is defined via

$$\sup_{\eta > 1} \mathrm{ess\,sup}\, \mathbb{E}_\eta \left[ (N - \eta)^+ \mid \mathcal{F}_\eta \right].$$

However, in the absence of further restrictions, studying this quantity for the problem at hand does not seem possible. In fact, as long as $\mathbb{P}^{\otimes \eta} := \mathbb{P} \otimes \cdots \otimes \mathbb{P}$ and $\mathbb{Q}^{\otimes \eta} := \mathbb{Q} \otimes \cdots \otimes \mathbb{Q}$ have a total variation distance smaller than 1, one can couple the given process and a process where all $X_t$'s are drawn from $\mathbb{Q}$ so that with non-zero probability the two processes have identical sequences. In this case, we either lose control of the null and $\alpha$ cannot be arbitrarily close to zero, or we maintain control over the null but with non-zero probability the detection delay is infinite.

# 6 Experiments

We collect our experiments on synthetic data in Section 6.1 and on the MNIST data set in Section 6.2. We refer to Appendix A.3 for additional experiments and a numerical comparison of different thresholds for the stopping rule. To interpret the change point detection performance of the proposed method, we compare its average run length (ARL) and expected detection delay (EDD) to the existing kernel-based methods presented in Table 1.[4] For all experiments, we use the Gaussian kernel $K(\mathbf{x}, \mathbf{y}) = e^{-\gamma \|\mathbf{x}-\mathbf{y}\|_2^2}$ with $\gamma$ set by the median heuristic [12] or its RFF approximation, depending on the algorithm. All results were obtained on a PC with Ubuntu 20.04 LTS, 124GB RAM, and 32 cores with 2GHz each.

## 6.1 Synthetic data

In this section, we evaluate the runtimes of different configurations of the proposed Online RFF-MMD algorithm and compare its change point detection performance on synthetic data to that of other kernel-based approaches.

**Runtime.** Figure 2 summarizes the runtime results of Algorithm 1 with the number of random Fourier features $r \in \{10, 50, 100, 500, 1\,000\}$ and for streams of length up to $n = 250\,000$. The experiments verify the $\mathcal{O}(r \log n)$ runtime complexity of the proposed algorithm, derived analytically in Section 4.2. We note that the dependence on $d$ is linear; we consider $d = 1$ only.

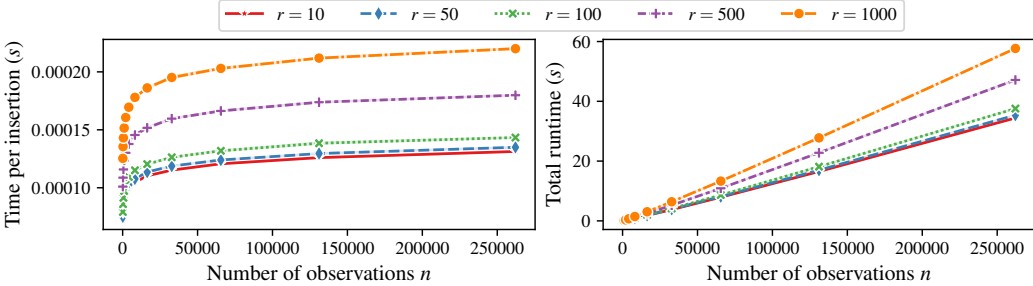

Figure 2: Average runtime (10 repetitions) of RFF-MMD per insert operation (left) and total (right).

**ARL vs. expected detection delay.** To illustrate the EDD for a given target ARL, we reproduce the experiments of Wei and Xie [59, Figure 4], also taking our method into account. Specifically, we consider the pre-change distribution $\mathbb{P} = \mathcal{N}(\mathbf{0}_{20}, \mathbf{I}_{20})$ and set the parameters of each algorithm as follows. Matching the settings of the reproduced experiment, we choose $B_{\max} = 50$ and $N = 15$ for online kernel CUSUM; for Scan B-statistics and NewMA, we set $B_0 = 50$. The remaining parameters of NewMA then follow from the heuristics detailed by the authors [24]. For Online RFF-MMD, we set $r = 1\,000$. We compute the thresholds for a given target ARL by processing $10 \times$ (target ARL) samples with each algorithm, repeating for 100 Monte Carlo (MC) iterations, and computing the $1 - 1/$(target ARL) quantile of the resulting test statistics. For approximating the EDD of each algorithm, we draw 64 samples from $\mathbb{P}$, respectively, before sampling from $\mathbb{Q}$; we report the average over 100 repetitions. OKCUSUM and ScanB additionally receive $1\,000$ samples from $\mathbb{P}$ upfront, to use as a reference sample. For NewMA, we process 400 additional samples from $\mathbb{P}$ for both the MC estimate and the EDD experiment, to reduce its variance.

Having processed the indicated number of samples from $\mathbb{P}$, we then start sampling from either a mixed normal, a Laplace, or a Uniform distribution. Figure 3 collects the average detection delay; the respective post-change distribution $\mathbb{Q}$ is given on top. The results show that our algorithm achieves a smaller detection delay than the competitors for all considered post-change distributions, sometimes by a large margin.

---

[4] All code replicating our experiments is available in the supplement and at `https://github.com/FlopsKa/rff-change-detection`.

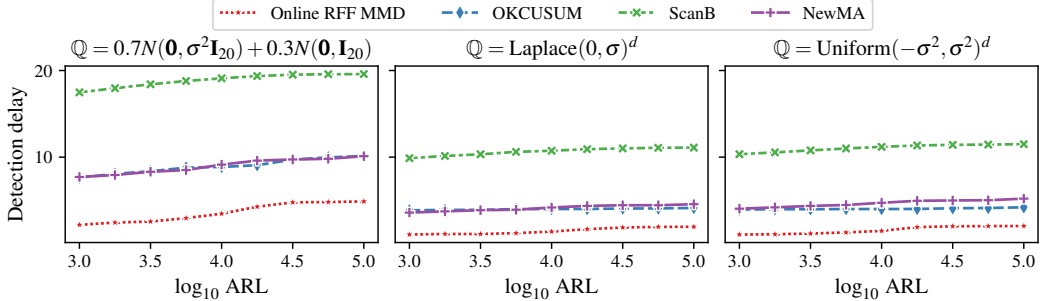

Figure 3: Average detection delay from $\mathbb{P} = \mathcal{N}(\mathbf{0}_{20}, \mathbf{I}_{20})$ to the $\mathbb{Q}$ indicated on top ($d = 20$, $\sigma = 2$).

## 6.2 MNIST data

In this section, we interpret the MNIST data set [9] as high-dimensional data stream, similar to Wei and Xie [59, Figure 7], with the goal of detecting a change when the digit changes from 0 to a different digit. The experimental setup matches that of Section 6.1, but with $d = 784$. Figure 4 collects our results; the results for the digits 4–6 are similar and in Appendix A.3.1. Similar to Figure 3, the proposed Online RFF-MMD algorithm shows very good performance. In particular, it achieves a lower EDD than all tested competitors throughout.

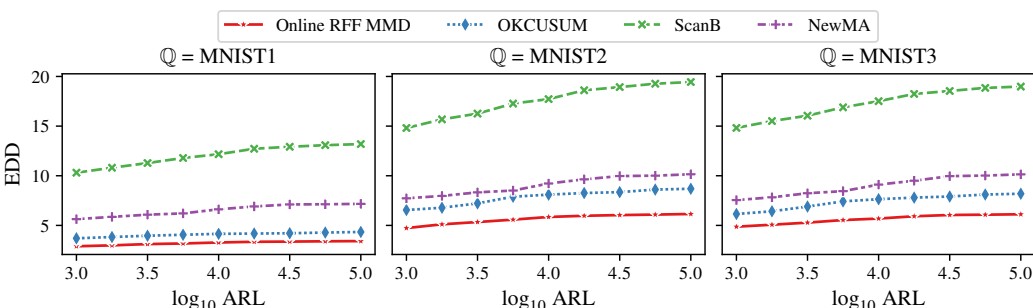

Figure 4: Average detection delay from MNIST digit 0 to digits 1, 2, and 3 (left to right).

## 7 Limitations

As with all kernel-based tests, the choice of kernel impacts the power of the test. While our theoretical guarantees (Section 4.3) hold for any kernel satisfying Assumption 1 and do not require any pre-change data, one usually selects the kernel or its parameters using a few available samples in practice. While kernel optimization is not the focus of this work, there exist works [20–22, 34, 50, 51, 16–18] to (approximately) achieve this goal; it is interesting future work to tackle this problem in the sequential setting. A separate future direction is considering non-i.i.d. data, for example, data that is strongly mixing [6] or data exhibiting functional dependence [60].

## Acknowledgments and Disclosure of Funding

The authors thank Zoltán Szabó and Tengyao Wang for helpful discussions. FK thanks Georg Gntuni and Marius Bohnert for exchanges on the algorithm's implementation. This work was supported by the pilot program Core-Informatics of the Helmholtz Association (HGF).

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

# A Appendix

This appendix is structured as follows. We detail the extension of Online RFF-MMD to take a known or observed pre-change distribution into account in Appendix A.1. In particular, we show that in these settings tighter thresholds are possible. Appendix A.2 shows how to adapt our algorithm to detect multiple change points and gives corresponding guarantees. Additional numerical results of MMD-RFF are in Appendix A.3; we include numerical results w.r.t. our tighter thresholds in Appendix A.3.5. Appendix A.4 collects all our proofs, with auxiliary results in Appendix A.5 and external statements in Appendix A.6.

## A.1 Known or estimable pre-change distribution

In this section, we discuss practical extensions of Online RFF-MMD when additional information about the pre-change distribution is available. Specifically, in Section A.1.1, we show how the algorithm can be adapted to settings in which (i) one has access to historical data known to be from the pre-change distribution, or (ii) one knows the pre-change distribution exactly. In Section A.1.2, we describe how this additional information allows sharpening the thresholds proposed in Theorems 1 and 2 in the main text.

### A.1.1 Incorporating information of the pre-change distribution

To begin with, we consider the setting in which, for some $\nu \in \mathbb{N}$, historical data $X_{-\nu+1}, \ldots, X_0$ known to be from the (nonetheless unknown) pre-change distribution $\mathbb{P}$ is available. This setting has been studied in the literature from both the parametric [44] and non-parametric [59] perspectives. It is straightforward to extend the Online RFF-MMD stopping time defined in the main text to take advantage of the additional information. Intuitively, for each local test one may prepend the historical data to the block of data taken to be from the pre-change distribution. More formally, the following stopping time can be used:

$$N = \inf \left\{ n \geq 2 \mid \bigcup_{j=0}^{\lfloor \log_2(n) \rfloor - 1} \sqrt{\frac{2^j(n+\nu-2^j)}{n+\nu}} \, \mathrm{MMD}_{\hat{K}} \left[ X_{(-\nu+1):(n-2^j)}, X_{(n-2^j+1):n} \right] > \lambda_n \right\}.$$

This stopping time can be implemented similarly to Algorithm 1, and such an implementation features the same time and space complexity as the original algorithm.

Next, we consider the setting in which the pre-change distribution $\mathbb{P}$ is known exactly. With this additional information, rather than performing local two sample tests, one may perform local one sample tests where the RFF approximation to the mean embedding of the data's empirical distribution is compared to the RFF approximation to the mean embedding of $\mathbb{P}$. More formally, the following stopping time can be used:

$$N = \inf \left\{ n \geq 2 \mid \bigcup_{j=0}^{\lfloor \log_2(n) \rfloor - 1} \sqrt{2^j} \, \mathrm{MMD}_{\hat{K}} \left[ \mathbb{P}, X_{(n-2^j+1):n} \right] > \lambda_n \right\}$$

The exact knowledge of $\mathbb{P}$ permits a precise approximation or the exact computation of $\mathbb{E}_{X \sim \mathbb{P}} \left[ \hat{z}_K(X) \mid \omega_1, \ldots, \omega_r \right]$, given $\omega_1, \ldots, \omega_r$ sampled from $\Lambda$. Again, this stopping time can be implemented similarly to Algorithm 1, enjoying the same time and space complexity as the original algorithm.

### A.1.2 Sharper thresholds through knowledge of the pre-change distribution

Access to additional information about the pre-change distribution paves the way to sharpening the thresholds proposed in Theorems 1 and 2. Indeed, although Theorem 4 suggests the thresholds proposed in the main paper are unimprovable up to constants, in practice these thresholds will be quite conservative as they are completely agnostic to the distribution of the data.

The main tool for proving Theorems 1 and 2 is Lemma 5, which controls the tail behavior of the tests performed by Online RFF-MMD under their respective local nulls. With additional knowledge of the pre-change distribution, Lemma 5 can be significantly sharpened by taking the second moment of the feature map into account. To that end, we have the following result.

**Lemma 1.** *Given two independent samples $\{X_1, \ldots, X_n\}$ and $\{Y_1, \ldots, Y_m\}$ each with mutually independent entries drawn from some $\mathbb{P} \in \mathcal{M}_1^+$, for any $\varepsilon > 0$, it holds that*

$$\mathbb{P}\big(\mathrm{MMD}_{\hat{K}}[X_{1:n}, Y_{1:m}] > \varepsilon\big) \leq 4\exp\left(-\frac{1}{2}\min(m,n)\varepsilon^2\left[\tilde{\sigma}^2 + 2\varepsilon\right]^{-1}\right),$$

*where $\tilde{\sigma}^2 = \mathbb{E}_{X\sim\mathbb{P}}\left[K\left(X, X\right)\right] - \mathbb{E}_{X,Y\sim\mathbb{P}}\left[K\left(X, Y\right)\right]$*

Lemma 1 allows the following improvements of Theorems 1 and 2.

**Corollary 1.** *For any $\gamma > 1$, replacing the thresholds in Theorem 1 with the scale dependent thresholds*

$$\lambda_{n,j} = \frac{2\sqrt{2}f^2(\gamma)}{\sqrt{\min(2^j, n - 2^j)}} + \tilde{\sigma}f(\gamma), \qquad n \in \mathbb{N}, \; j \leq \log_2(n) \tag{13}$$

$$f(\gamma) = \sqrt{2\log(16\gamma\log_2(2\gamma))}$$

*it holds that $\mathbb{E}_\infty[N] \geq \gamma$ where $N$ is as defined in (8).*

**Corollary 2.** *For any $\alpha \in (0, 1)$, replacing the thresholds in Theorem 2 with the scale dependent thresholds*

$$\lambda_{n,j} = \frac{2\sqrt{2}f^2(\alpha, n)}{\sqrt{\min(2^j, n - 2^j)}} + \tilde{\sigma}f(\alpha, n), \qquad n \in \mathbb{N}, j \leq \log_2(n)$$

$$f(\alpha, n) = \sqrt{2(\log(4n/\alpha) + 2\log(\log_2(n)) + \log(\log_2(2n)))}$$

*it holds that $\mathbb{P}_\infty(N < \infty) \leq \alpha$ where $N$ is as defined in (8).*

To put the above results in context, we detail Corollary 1. On large scales (large $j$-s), where detections tend to occur, the scale dependent thresholds behave approximately like $\tilde{\sigma}\sqrt{2\log(4\gamma\log_2(2\gamma))}$, which differs from the threshold used in Theorem 1 by a factor of $\tilde{\sigma}$. Therefore, when $\tilde{\sigma}$ is significantly smaller than 1, the thresholds in (13) will be significantly smaller than those suggested by Theorem 1. As the kernel is assumed to be bounded from above by 1, these smaller thresholds generally occur.

## A.2 Online detection of multiple change points

In this section, we explain how the algorithm in the main part can be extended to detect multiple change points in an online fashion. Indeed, consider the setting where data $X_1, X_2, \ldots$ are observed sequentially. There is a sequence of integers $(\eta_k)_{k\in\mathbb{N}}$ with $\eta_0 = 0$ and $\eta_k < \eta_{k+1}$ for each $k \in \mathbb{N}$, and a sequence of measures $(\mathbb{Q}_k)_{k\in\mathbb{N}}$ each in $\mathcal{M}_+^1$ with $\mathbb{Q}_k \neq \mathbb{Q}_{k+1}$ for each $k \in \mathbb{N}$ such that for each $t \in \mathbb{N}$ the data are distributed according to

$$X_t \sim \begin{cases} \mathbb{Q}_1 & \text{if} \quad \eta_0 < t \leq \eta_1 \\ \mathbb{Q}_2 & \text{if} \quad \eta_1 < t \leq \eta_2 \\ \vdots & \end{cases}.$$

Consider running Algorithm 1 on this data stream using the threshold sequence given in Theorem 2. As soon as a change is detected, say at location $n$, we re-start the algorithm at $X_{n+1}$. However, the index on the threshold sequence is maintained. More precisely: having detected a change at time $n$, at the $(n + j)$-th iteration (for $j \geq 1$), we construct local statistics over the interval $\{n + 1, \ldots, n + j\}$ according to Section 4.1, however the local statistics continue to be compared with the threshold $\lambda_{n+j}$. We are able to prove the following guarantee on the change point locations recovered in this way:

**Theorem 5.** *For any deterministic time $T < \infty$ let $M$ denote the number of $\eta_k$-s taking values in $\{1, \ldots, T\}$. Write $\delta_k = \min(\eta_k - \eta_{k-1}, \eta_{k+1} - \eta_k)$ for $k = 1, \ldots, M - 1$ and $\delta_M = \min(\eta_M - \eta_{M-1}, T - \eta_M)$ for the effective sample size associated with the $k$-th change point location. For some $\alpha \in (0, 1)$, let $\hat{M}$ be the number of change points detected up to time $T$ using the threshold sequence given in Theorem 2 and the procedure described above, and let $\hat{\eta}_1, \ldots, \hat{\eta}_{\hat{M}}$ be their locations. If $\cup_{k=1}^M \mathrm{supp}(\mathbb{Q}_k) \subseteq \mathcal{X}$ for some compact set $\mathcal{X} \subset \mathbb{R}^d$,*

$$\delta_k \geq C_1 \frac{\log(2T/\alpha)}{(\mathrm{MMD}_K[\mathbb{Q}_k, \mathbb{Q}_{k+1}])^2} \quad \text{for} \quad k = 1, \ldots, M,$$

*and the number of random Fourier features is chosen so that*

$$\sqrt{r} \geq C_2 \frac{h(d, |\mathcal{X}|, \sigma) + \sqrt{2\log(2/\alpha)}}{\left(\min_{k=1,\dots M} \mathrm{MMD}_K[\mathbb{Q}_k, \mathbb{Q}_{k+1}]\right)^2},$$

*then with probability at least $1 - 2\alpha$ it holds that $\hat{M} = M$ and*

$$\eta_k < \hat{\eta}_k \leq \eta_k + C_3 \frac{\log(2T/\alpha)}{\left(\mathrm{MMD}_K[\mathbb{Q}_k, \mathbb{Q}_{k+1}]\right)^2} \quad for \quad k = 1, \dots M.$$

*Here $h(d, |\mathcal{X}|, \sigma)$ is as given in Theorem 3, and $C_1$, $C_2$, and $C_3$ are absolute constants independent of $\alpha$ and the sequences $\{\delta_k\}_{k=1,\dots,M}$ and $\{\mathrm{MMD}_K[\mathbb{Q}_k, \mathbb{Q}_{k+1}]\}_{k=1,\dots,M}$.*

### A.3    Additional experiments

In this section, we summarize additional numerical results. The MNIST results with the MC threshold as in the main text are in Section A.3.1. We include results obtained without threshold estimation in Section A.3.4. A comparison of the distribution dependent and distribution-free bounds is in Section A.3.5. Numerical studies on the Human Activity Sensing Consortium (HASC) and on the MarzukaBL data sets are in Section A.3.2 and in Section A.3.3, respectively.

### A.3.1    MNIST data digits 4–9

In the following Figure 5, we collect the change detection performances of the algorithms of Table 1 on MNIST data, where the digit changes from 0 to one of 4–9; the results on the other digits and the experimental setup are in Section 6.2. As in the corresponding experiment in the main part of this article, our proposed Online RFF-MMD algorithm consistently achieves the lowest expected detection delay, highlighting its good practical performance.

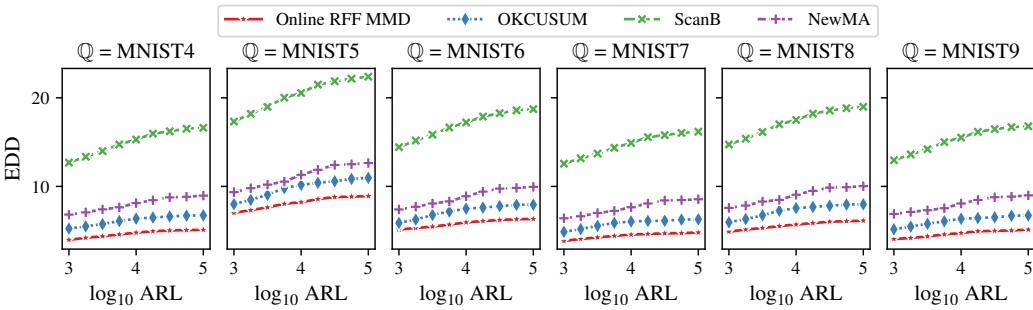

Figure 5: Average detection delay from MNIST digit 0 to digits 4–9 (left to right).

### A.3.2    HASC (Human Activity Sensing Consortium) data

This section collects our experiments on the Human Activity Sensing Consortium (HASC; available at http://hasc.jp/hc2011/) challenge 2011 data set, which is also considered in Liu et al. [35], Li et al. [33], Wei and Xie [59]. As in Wei and Xie [59], we consider the change from *walking* to *staying* of participant 101. For preprocessing, we order the corresponding csv-files in the data set lexicographically, omitting the first 1 596 (detailed below) samples of *walking*, and then concatenating 100 *walking* observations and 100 *staying* observations to obtain a total of 10 data sets (with $d = 3$) with a single change point each. For obtaining the thresholds for the proposed Online RFF MMD, NewMA, Scan-B statistics, and OKCUSUM change detection algorithms, we proceed as elaborated in Section 6.1, that is, we use a Monte Carlo approach with $\alpha = 0.1$ on all 10 *walking* data sets. As per Wei and Xie [59], for ScanB and OKCUSUM, we set the number of windows $N = 14$ and the window length $w = 114$, implying that both algorithms receive $14 \cdot 114 = 1\,596$ samples from *walking* upfront. Likewise, we process 1 596 elements with NewMA upfront. All kernel-based approaches use the (approximated) Gaussian kernel with the $\gamma > 0$ parameter set by the median heuristic.

For the density ratio-based (i.e., non-kernel-based) RuLSIF algorithm—which showed the best performance on HASC in Liu et al. [35]—, we use the python `changepoynt` implementation and consider the $l_2$-norm of each three-dimensional observation.[5] We then obtain the change scores for the full data set and select the point with the highest score as predicted change point, as done in Liu et al. [35, Section 4.2]. To match their setup, we set the window length to $50$, the number of windows to $10$, and $\alpha = 0.1$. Additionally, due to the large number of "too early" cases reported in this setup, we introduce an offset and take the window length to be equal to the offset. In this setup, we consider the point with the highest score plus the offset as reported change point. A "miss" can not occur, due to the scores reported to RuLSIF having at least one maximum. Table 2 collects our results.

Table 2: Comparison of change detection algorithms on 10 data sets derived from the HASC data set. "Too early" refers to the number of times an algorithm reported a change point before the actual change occurred. "Miss" reports cases in which no change was reported.

| Algorithm | Average delay | Too early | Miss |
|---|---|---|---|
| Online RFF MMD | 21.86 | 2 | 1 |
| NewMA | 34.25 | 1 | 5 |
| ScanB | 31.20 | 0 | 0 |
| OKCUSUM | 17.44 | 1 | 0 |
| RuLSIF | 4.50 | 8 | 0 |
| RuLSIF (offset $= 30$) | 20.38 | 2 | 0 |
| RuLSIF (offset $= 40$) | 35.0 | 3 | 0 |
| RuLSIF (offset $= 50$) | 39.2 | 0 | 0 |

We emphasize that in the above case all algorithms except for the proposed one and RuLSIF receive $1\,596$ samples upfront, to use as a reference sample. Even though this fundamentally favors the kernel-based competitors, our proposed method achieves results that are comparable to those of the other kernel-based approaches and to RuLSIF.

### A.3.3 MazurkaBL data

In Kosta et al. [26], the authors found that change points in loudness information of Chopin's Mazurkas correspond to score positions having dynamic markings, tempo, or expression markings, among others. To further validate our algorithm's performance, we additionally run the proposed method on the "M17-4" sample (illustrated in Figure 6) of pianist "pid50534-05" of their MazurkaBL [27] data set with the goal of detecting these changes. Here, the dimensionality is $d = 1$.

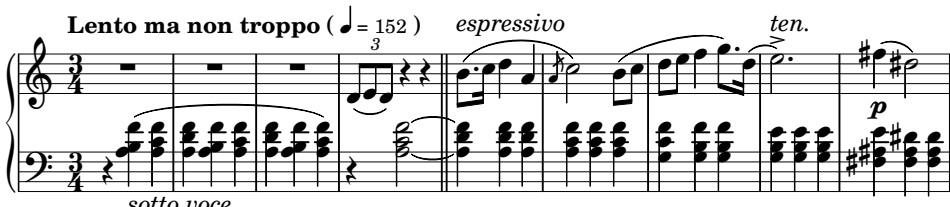

Figure 6: An excerpt of Frédéric Chopin's Mazurka Op. 17 No. 4.

As in the main article (see Section 6), we approximate the Gaussian kernel with $\gamma > 0$ set by the median heuristic. For obtaining the thresholds using Monte Carlo iterations, we slice the data along each annotated change point, where we consider the annotations provided by Kosta et al. [27] as ground truth, and compute the test statistics obtained by the proposed Online RFF MMD algorithm individually on each one. We then select the $1 - 0.1$-quantile across all test statistics so obtained as threshold.

To detect change points, we again slice the loudness data, but now such that each slice contains precisely one change point, which yields a total of 25 samples with an average length of $1\,561.32$.

---

[5]The `changepoynt` library is available at `https://github.com/Lucew/changepoynt`.

We process each one of these with our proposed method and consider the first time the test statistic exceeds the threshold as "change". In total, our proposed method flags 10 change points too early, and, on the remaining 15 has an average detection delay of 73.67, with a median detection delay of 64.0.

While, when contrasting these results with the results obtained on the MNIST (Section A.3) and HASC (Section A.3.2) data sets, detecting changes in the selected loudness data of a Mazurka seems substantially more challenging, the proposed algorithm still manages to detect many changes with a relatively small delay.

### A.3.4  Distribution-free bound

In this section, we show the change detection performance of the proposed Online RFF-MMD algorithm if no pre-change sample is used to estimate the threshold. Instead, we use Theorem 1 to compute the distribution-free threshold sequence $\{\lambda_n \mid n \in \mathbb{N}\}$ for a given target ARL. To obtain an EDD estimate, we sample and process $512$ observations from MNIST digit $0$ (pre-change) and $1\,024$ samples from digits $1$–$9$ (post-change), respectively, averaging the detection delay over $100$ repetitions. The results are in Figure 7. When comparing to Figure 4 and Figure 5, the figure shows that our method has an increased detection delay, which is due to the looser distribution-free bound (see Section A.3.5 for a numerical comparison). Still, except for the change to the digit $5$ with a guaranteed ARL of $10^5$, Online RFF-MMD detects all changes reliably.

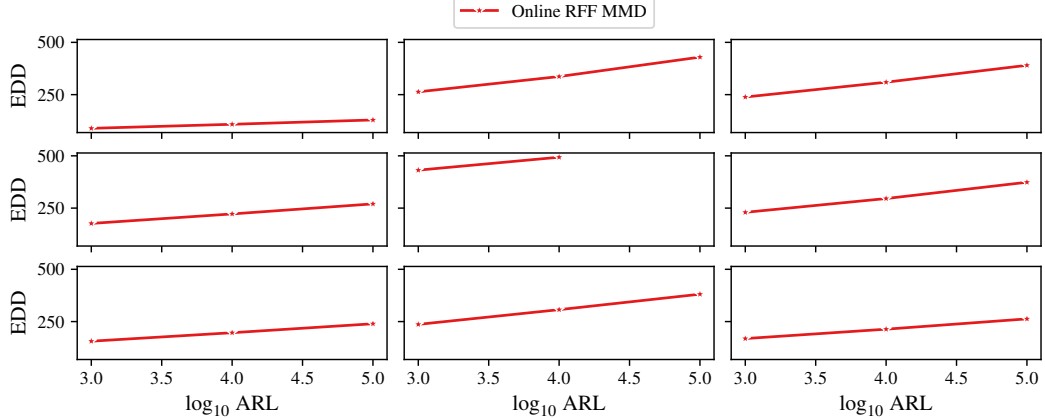

Figure 7: Average detection delay from MNIST digit 0 to digits 1–3, 4–6, 7–9 (top to bottom) with the distribution-free threshold sequence of Theorem 1.

### A.3.5  Threshold comparison

In this section, we compare the tightness of our thresholds in the offline two-sample testing setting. Specifically, we fix the level $\alpha = 0.01$ and let $\mathbb{P} = \mathbb{Q} = \mathcal{N}(0,1)$. We then approximate the $1 - \alpha$ quantile of $\mathrm{MMD}_{\hat{K}}\left(\hat{\mathbb{P}}_n, \hat{\mathbb{Q}}_n\right)$ (with $n = 1\,000$, $\hat{K}$ approximating the Gaussian kernel with $r = 1\,000$ RFFs, and $\gamma > 0$ set by the median heuristic) by (i) obtaining new samples from $\mathbb{P}, \mathbb{Q}$ and (ii) permuting a fixed sample from $\mathbb{P}, \mathbb{Q}$ for $1\,000$ rounds. Figure 8 shows the respective histograms and the estimated quantiles along with the thresholds obtained by Lemma 1 and Lemma 5, respectively. As one expects, the figure shows that the variance estimate used in Lemma 1 allows to obtain a tighter bound, where we consider the resampling/permutation-based thresholds as ground truth. We emphasize that independent of the threshold used, the resulting test is consistent against any fixed alternative.

### A.4  Proofs

This section is dedicated to our proofs. The proof of Theorem 1 is in Appendix A.4.1, that of Theorem 2 is in Appendix A.4.2, that of Theorem 3 is in Appendix A.4.3, and that of our minimax result (Theorem 4) is in Section A.4.4. The tighter threshold detailed in Lemma 1 is proved in

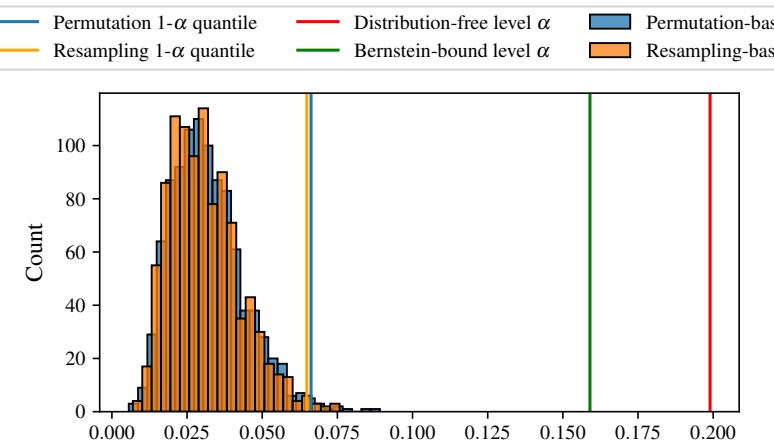

Figure 8: Comparison of different thresholds for the acceptance region of the MMD two-sample test.

Appendix A.4.5 and we state the proof for our multiple change point detection result (Theorem 5) in Appendix A.4.6.

### A.4.1 Proof of Theorem 1

*Proof.* For ease of reading, for each $n \geq 2$ and $j = 0, \ldots, \lfloor \log_2(n) \rfloor - 1$, put

$$\hat{M}_{n,j} := \sqrt{\frac{2^j(n - 2^j)}{n}} \, \mathrm{MMD}_{\hat{K}} \left[ X_{1:(n-2^j)}, X_{(n-2^j+1):n} \right]. \tag{14}$$

By the law of total expectation, we have that

$$\begin{aligned}
\mathbb{E}_\infty[N] &= \mathbb{E}_\infty[N \mid N \leq 2\gamma] \, \mathbb{P}_\infty(N \leq 2\gamma) + \mathbb{E}_\infty[N \mid N > 2\gamma] \, \mathbb{P}_\infty(N > 2\gamma) \\
&\geq 2\gamma \left[1 - \mathbb{P}_\infty(N \leq 2\gamma)\right].
\end{aligned} \tag{15}$$

Putting $\lambda = \sqrt{2\log(4\gamma \log_2(2\gamma))}$, a union bound argument together with Lemma 5 gives that

$$\begin{aligned}
\mathbb{P}_\infty(N \leq 2\gamma) &= \mathbb{P}_\infty \left( \cup_{n=2}^{2\gamma} \cup_{j=0}^{\lfloor \log_2(n) \rfloor - 1} \hat{M}_{n,j} > \sqrt{2} + \lambda \right) \\
&\leq \sum_{n=1}^{2\gamma} \sum_{j=0}^{\lfloor \log_2(n) \rfloor - 1} \int \cdots \int \mathbb{P}_\infty \left( \hat{M}_{n,j} > \sqrt{2} + \lambda_n \mid \omega_1, \ldots, \omega_r \right) \mathrm{d}\Lambda(\omega_1) \ldots \mathrm{d}\Lambda(\omega_r) \\
&\leq 2\gamma \log_2(2\gamma) e^{-\lambda^2/2} = \frac{1}{2}.
\end{aligned} \tag{16}$$

Finally, plugging (16) into (15) proves the desired result. $\square$

### A.4.2 Proof of Theorem 2

*Proof.* Write

$$\pi_n = \sqrt{2 \left(\log(n/\alpha) + 2\log(\log_2(n)) + \log(\log_2(2n))\right)} \tag{17}$$

for each $n \in \mathbb{N}$. Applying standard peeling arguments [64, 57], we have that

$$\begin{aligned}
\mathbb{P}_\infty(N < \infty) &= \mathbb{P}_\infty \left( \cup_{n=2}^{\infty} \cup_{j=0}^{\lfloor \log_2(n) \rfloor - 1} \hat{M}_{n,j} > \sqrt{2} + \pi_n \right) \\
&\leq \sum_{l=1}^{\infty} \mathbb{P}_\infty \left( \cup_{n=2^l}^{2^{l+1}} \cup_{j=0}^{\lfloor \log_2(n) \rfloor - 1} \hat{M}_{n,j} > \sqrt{2} + \pi_n \right) \\
&\leq \sum_{l=1}^{\infty} 2^l \max_{2^l \leq n \leq 2^{l+1}} \mathbb{P}_\infty \left( \cup_{j=0}^{\lfloor \log_2(n) \rfloor - 1} \hat{M}_{n,j} > \sqrt{2} + \pi_n \right) \tag{18a} \\
&\leq \sum_{l=1}^{\infty} 2^l \max_{2^l \leq n \leq 2^{l+1}} \sum_{j=0}^{\lfloor \log_2(n) \rfloor - 1} \mathbb{P}_\infty \left( \hat{M}_{n,j} > \sqrt{2} + \pi_n \right), \tag{18b}
\end{aligned}$$

where in line (18a), we apply a union bound and bound the resulting sum by its maximum. Then, applying Lemma 5 as was done in (16), we obtain that

$$\mathbb{P}_\infty(N < \infty)$$

$$\leq \sum_{l=1}^\infty 2^l \max_{2^l \leq n \leq 2^{l+1}} \sum_{j=0}^{\lfloor \log_2(n) \rfloor - 1} \int \cdots \int \mathbb{P}_\infty \left( \hat{M}_{n,j} > \sqrt{2} + \pi_n \mid \omega_1, \ldots, \omega_r \right) d\Lambda(\omega_1) \ldots d\Lambda(\omega_r)$$

$$\leq \sum_{l=1}^\infty l 2^l e^{-(\pi_{2^l})^2/2}.$$

Finally using the facts that (i) $\exp\left(-\pi_{2^l}^2/2\right) \leq \alpha l^{-2}(l+1)^{-1} 2^{-l}$ for all $l \in \mathbb{N}$ and (ii) $\sum_{l=1}^\infty l^{-1}(l+1)^{-1} = 1$, we obtain that $\mathbb{P}_\infty(N < \infty) \leq \alpha$. $\qquad\square$

### A.4.3 Proof of Theorem 3

*Proof.* We first observe that for any triplet of integers $(m, n, \nu)$ satisfying (i) $m \leq n$ and (ii) $\nu \leq n/2$, given two samples

$$X_{1:n} = \left\{ X_1, \ldots, X_{n-\nu}, \tilde{Y}_{n-\nu+1}, \ldots, \tilde{Y}_n \right\} \qquad \text{and} \qquad Y_{1:m} = \{Y_1, \ldots, Y_m\},$$

with mutually independent entries taking values in some bounded set $\mathcal{X} \subset \mathbb{R}^d$ where the $X$'s are sampled from $\mathbb{P}$ and the $Y$'s and $\tilde{Y}$'s are sampled from $\mathbb{Q}$ for some $\mathbb{P}, \mathbb{Q} \in \mathcal{M}_1^+(\mathcal{X})$ not identical, for any $\varepsilon > 0$, it holds that

$$\mathbb{P}\left( \sqrt{\frac{nm}{n+m}} \, \mathrm{MMD}_{\hat{K}}[X_{1:n}, Y_{1:m}] \leq \sqrt{2} + \varepsilon \right)$$

$$\leq \mathbb{P}\left( 2\sqrt{m \sup_{\mathbf{x}, \mathbf{y} \in \mathcal{X}} \left| \hat{K}(\mathbf{x}, \mathbf{y}) - K(\mathbf{x}, \mathbf{y}) \right|} > \frac{1}{3}\left[ \sqrt{m}\left( \frac{\sqrt{2}-1}{2\sqrt{2}} \right) \mathrm{MMD}_K[\mathbb{P}, \mathbb{Q}] - \left( \varepsilon + \frac{10}{\sqrt{2}} \right) \right] \right)$$

$$+ 4\exp\left( -\frac{1}{18}\left\{ \left[ \sqrt{m}\left( \frac{\sqrt{2}-1}{2\sqrt{2}} \right) \mathrm{MMD}_K[\mathbb{P}, \mathbb{Q}] - \left( \varepsilon + \frac{10}{\sqrt{2}} \right) \right] \vee 0 \right\}^2 \right) \qquad (19)$$

where $\mathrm{MMD}_K$ is as in (4) and $\mathrm{MMD}_{\hat{K}}$ is as defined in (7). To show (19), let the $\tilde{X}$'s below be sampled independently from $\mathbb{P}$ and introduce the quantities

$$\Delta_1 = \left\| \frac{1}{n}\left[ \sum_{i=1}^{n-\nu} K(\cdot, X_i) + \sum_{i=n-\nu+1}^n K(\cdot, \tilde{X}_i) \right] - \frac{1}{m}\sum_{i=1}^m K(\cdot, Y_i) \right\|_{\mathcal{H}_K}$$

$$\Delta_2 = \left\| \frac{1}{\nu}\sum_{i=n-\nu+1}^n K(\cdot, \tilde{X}_i) - \frac{1}{\nu}\sum_{i=n-\nu+1}^n K(\cdot, \tilde{Y}_i) \right\|_{\mathcal{H}_K}.$$

Note that by the reverse triangle inequality

$$\mathrm{MMD}_K[X_{1:n}, Y_{1:m}] \geq \Delta_1 - \sqrt{\frac{\nu}{n}}\Delta_2.$$

Consequently, using the above and by repeated applications of the triangle inequality, one has that

$$\sqrt{\frac{nm}{n+m}}\,\mathrm{MMD}_{\hat{K}}[X_{1:n}, Y_{1:m}]$$

$$\geq \sqrt{\frac{nm}{n+m}}\left( 1 - \sqrt{\frac{\nu}{n}} \right)\mathrm{MMD}_K[\mathbb{P}, \mathbb{Q}]$$

$$- \sqrt{\frac{nm}{n+m}}\left| \mathrm{MMD}_{\hat{K}}[X_{1:n}, Y_{1:m}] - \mathrm{MMD}_K[X_{1:n}, Y_{1:m}] \right| \qquad (20\mathrm{a})$$

$$- \sqrt{\frac{nm}{n+m}}\left| \Delta_1 - \mathbb{E}[\Delta_1] \right| - \sqrt{\frac{2m}{n+m}}\sqrt{\frac{\nu}{2}}\left| \Delta_2 - \mathbb{E}[\Delta_2] \right| \qquad (20\mathrm{b})$$

$$- \sqrt{\frac{nm}{n+m}}\left| \mathbb{E}[\Delta_1] - \mathrm{MMD}_K[\mathbb{P}, \mathbb{Q}] \right| - \sqrt{\frac{2m}{n+m}}\sqrt{\frac{\nu}{2}}\left| \mathbb{E}[\Delta_2] - \mathrm{MMD}_K[\mathbb{P}, \mathbb{Q}] \right|. \qquad (20\mathrm{c})$$

For term (20a), applying Lemmas 2 and 3 together with the fact that the $X$'s and $Y$'s take values in some compact $\mathcal{X} \subset \mathbb{R}^d$, we obtain that

$$\sqrt{\frac{nm}{n+m}} \left| \text{MMD}_{\hat{K}}\left[X_{1:n}, Y_{1:m}\right] - \text{MMD}_K\left[X_{1:n}, Y_{1:m}\right] \right| \leq 2\sqrt{m \sup_{\mathbf{x},\mathbf{y} \in \mathcal{X}} \left| \hat{K}\left(\mathbf{x}, \mathbf{y}\right) - K\left(\mathbf{x}, \mathbf{y}\right) \right|}. \tag{21}$$

For the penultimate term in (20c), applying the bound

$$\mathbb{E}\left|\text{MMD}_K\left[X_{1:n}, Y_{1:m}\right] - \text{MMD}_K\left[\mathbb{P}, \mathbb{Q}\right]\right| \leq 2\left(\frac{1}{\sqrt{n}} + \frac{1}{\sqrt{m}}\right) \tag{22}$$

for $X$'s sampled from $\mathbb{P}$ and $Y$'s sampled from $\mathbb{Q}$, whose proof can be found for instance in Section A.2 of [14], together with the bound $\sqrt{x+y} \geq \left(\sqrt{2}/2\right)\left(\sqrt{x} + \sqrt{y}\right)$ for all $x, y \geq 0$, which holds due to the concavity of the square root, one has that

$$\sqrt{\frac{nm}{n+m}} \left|\mathbb{E}\left[\Delta_1\right] - \text{MMD}_K\left[\mathbb{P}, \mathbb{Q}\right]\right| \leq \sqrt{\frac{nm}{n+m}} \mathbb{E}\left|\Delta_1 - \text{MMD}_K\left[\mathbb{P}, \mathbb{Q}\right]\right|$$
$$\leq 2\frac{\left(\sqrt{n} + \sqrt{m}\right)}{\sqrt{n+m}} \leq \frac{4}{\sqrt{2}}. \tag{23}$$

Identical arguments together with the fact that $m \leq n$ implies that $(2m)/(m+n) \leq 1$ give

$$(20b) \leq \sqrt{\frac{2m}{n+m}} \sqrt{\frac{\nu}{2}} \mathbb{E}\left|\Delta_2 - \text{MMD}_K\left[\mathbb{P}, \mathbb{Q}\right]\right| \leq \frac{4}{\sqrt{2}}. \tag{24}$$

Therefore, combining (21), (23), and (24), rearranging, and applying the rough bound

$$\mathbb{P}\left(\sum_{j=1}^{K} Z_j > x\right) \leq \sum_{j=1}^{K} \mathbb{P}\left(Z_j > x/K\right)$$

which holds for any $x \in \mathbb{R}$, $K \in \mathbb{N}$ and any random variables $Z_1, \ldots, Z_K$, we obtain that

$$(19) \leq \mathbb{P}\left(2\sqrt{m \times \sup_{\mathbf{x},\mathbf{y} \in \mathcal{X}} \left|\hat{K}\left(\mathbf{x}, \mathbf{y}\right) - K\left(\mathbf{x}, \mathbf{y}\right)\right|} > \right.$$

$$\left. \frac{1}{3}\left[\sqrt{m}\left(\frac{\sqrt{2}-1}{2\sqrt{2}}\right)\text{MMD}_K\left[\mathbb{P}, \mathbb{Q}\right] - \left(\varepsilon + \frac{10}{\sqrt{2}}\right)\right] \vee 0\right)$$

$$+ \mathbb{P}\left(\sqrt{\frac{nm}{n+m}} \left|\Delta_1 - \mathbb{E}\left[\Delta_1\right]\right| > \frac{1}{3}\left[\sqrt{m}\left(\frac{\sqrt{2}-1}{2\sqrt{2}}\right)\text{MMD}_K\left[\mathbb{P}, \mathbb{Q}\right] - \left(\varepsilon + \frac{10}{\sqrt{2}}\right)\right] \vee 0\right)$$
$$\tag{25a}$$

$$+ \mathbb{P}\left(\sqrt{\frac{\nu}{2}} \left|\Delta_2 - \mathbb{E}\left[\Delta_2\right]\right| > \frac{1}{3}\left[\sqrt{m}\left(\frac{\sqrt{2}-1}{2\sqrt{2}}\right)\text{MMD}_K\left[\mathbb{P}, \mathbb{Q}\right] - \left(\varepsilon + \frac{10}{\sqrt{2}}\right)\right] \vee 0\right). \tag{25b}$$

Note that we assume $0 \leq K(\cdot, \cdot) \leq 1$. Therefore, arguing as in Lemma 4, one can show that MMD as defined in (5) has the bounded differences property with constants (41). Hence, applying Theorem 6 to terms (25a) and (25b) one arrives at (19). Turning to the problem of interest we first make explicit the constants in Theorem 3:

$$C_1 = 2 \times C_3$$
$$C_2^{-1} = \frac{1}{3}\left[\left(\frac{\sqrt{2}-1}{4} - \frac{\sqrt{50}+\sqrt{6}}{\sqrt{C_3}}\right)\right]$$
$$C_3 = \left(\sqrt{6} + \sqrt{50} + \sqrt{54}\right)^2 \times \left(4/(\sqrt{2}-1)\right)^2.$$

Next, define the quantities

$$k^* = \min\left\{k \in \mathbb{N} \mid k \leq \eta \text{ and } \sqrt{k} \times \text{MMD}_K\left[\mathbb{P}, \mathbb{Q}\right] \geq \sqrt{C_3 \log\left(2\eta/\alpha\right)}\right\} \tag{26a}$$
$$t_{k^*} = \min\left\{t = 2^j \mid j \in \mathbb{N} \text{ and } t \leq k^*\right\}. \tag{26b}$$

Note that condition (9) guarantees that (26a) exists and it can be checked that $C_2 > 0$. Let $j^*$ be the $j$ appearing in (26b). Consequently, using (19) and the fact that $k^*/2 \leq t_{k^*} \leq k^*$, we obtain that

$$\mathbb{P}\left((N - \eta)^+ > k^*\right)$$

$$\leq \mathbb{P}\left(\bigcap_{j=0}^{\lfloor \log_2(\eta+k^*)\rfloor - 1} \sqrt{\frac{2^j(\eta + k^* - 2^j)}{\eta + k^*}}\, \mathrm{MMD}_{\hat{K}}\left[X_{I_{j^*,k^*}}, X_{J_{j^*,k^*}}\right] \leq \sqrt{2} + \lambda_{\eta+k^*}\right)$$

$$\leq \mathbb{P}\left(\sqrt{\frac{t_{k^*}(\eta + k^* - t_{k^*})}{\eta + k^*}}\, \mathrm{MMD}_{\hat{K}}\left[X_{I_{j^*,k^*}}, X_{J_{j^*,k^*}}\right] \leq \sqrt{2} + \lambda_{2\eta}\right)$$

$$\leq \mathbb{P}\left(2\sqrt{k^* \sup_{x,y \in \mathcal{X}} \left|\hat{K}(\mathbf{x},\mathbf{y}) - K(\mathbf{x},\mathbf{y})\right|} > \frac{1}{3}\left[\sqrt{k^*}\frac{\sqrt{2}-1}{4}\, \mathrm{MMD}_K[\mathbb{P},\mathbb{Q}] - \left(\lambda_{2\eta} + \frac{10}{\sqrt{2}}\right)\right]\right)$$

$$\tag{27a}$$

$$+ 4\exp\left(-\frac{1}{18}\left\{\left[\sqrt{k^*}\frac{\sqrt{2}-1}{4}\, \mathrm{MMD}_K[\mathbb{P},\mathbb{Q}] - \left(\lambda_{2\eta} + \frac{10}{\sqrt{2}}\right)\right] \vee 0\right\}^2\right). \tag{27b}$$

where for typographical reasons we have put:

$$I_{j^*,k^*} := 1 : (\eta + k^* - 2^{j^*})$$
$$J_{j^*,k^*} := (\eta + k^* - 2^{j^*} + 1) : (\eta + k^*),$$

for $j = 0, \ldots, \lfloor \log_2(\eta + k^*)\rfloor - 1$. Now (26a) together with the fact that $\lambda_{2\eta} + \frac{10}{\sqrt{2}} \leq (\sqrt{50} + \sqrt{6}) \times \sqrt{\log(2\eta/\alpha)}$ for all $\alpha \in (0,1)$ and $\eta \in \mathbb{N}$ guarantees that the term on the right of the inequality in (27a) is no larger than $C_2 \times \sqrt{k^*}\, \mathrm{MMD}_K[\mathbb{P},\mathbb{Q}]$. Hence, appealing to (10) and Theorem 7, we obtain that $(27a) \leq \alpha/2$. Moreover, since for each $k \leq \eta$ it holds that

$$\left\{\sqrt{k}\left(\frac{\sqrt{2}-1}{4}\right)\mathrm{MMD}_K[\mathbb{P},\mathbb{Q}] - \left(\lambda_{2\eta} + \frac{10}{\sqrt{2}}\right) \geq \sqrt{18 \times 3 \log\left(\frac{2\eta}{\alpha}\right)}\right\}$$

$$\subseteq \left\{\sqrt{k} \times \mathrm{MMD}_K[\mathbb{P},\mathbb{Q}] \geq \sqrt{C_3 \log\left(\frac{2\eta}{\alpha}\right)}\right\}$$

with $k^*$ defined as in (26a), we obtain that $(27b) \leq 4 \times (\alpha/2\eta)^3 \leq \alpha/2$. With these facts in place the theorem is proved. $\qquad \square$

### A.4.4   Proof of Theorem 4

*Proof.* Let $\mathbf{1} = (1, \ldots, 1)^\top \in \mathbb{R}^d$ and $\mathbf{0} = (0, \ldots, 0)^\top \in \mathbb{R}^d$, let $\delta_{\boldsymbol{x}}$ denote the Dirac measure (for any set $A \in \sigma(\mathbb{R}^d)$ and any $\boldsymbol{x} \in \mathbb{R}^d$, $\delta_{\boldsymbol{x}}(A) = 1$ if $\boldsymbol{x} \in A$ and 0 otherwise), and let

$$\mathcal{M}^* = \left\{\mathbb{P}, \mathbb{Q} \in \mathcal{M}_1^+\left(\mathbb{R}^d\right) \mid \mathbb{P} = p\delta_{\mathbf{1}} + (1-p)\delta_{\mathbf{0}}, \mathbb{Q} = q\delta_{\mathbf{1}} + (1-q)\delta_{\mathbf{0}},\right.$$
$$\left. p, q \in [1/4, 3/4] \text{ and } q - p \geq 1/4\right\}.$$

Therefore, for any $\mathbb{P}, \mathbb{Q} \in \mathcal{M}^*$, making use of the symmetry of $K$, we have that

$$(\mathrm{MMD}_K[\mathbb{P},\mathbb{Q}])^2$$
$$= \mathbb{E}_{X,X'\sim\mathbb{P}}[K(X,X')] + \mathbb{E}_{Y,Y'\sim\mathbb{Q}}[K(Y,Y')] - 2\mathbb{E}_{X\sim\mathbb{P},Y\sim\mathbb{Q}}[K(X,Y)] \tag{28}$$
$$= p^2 K(\mathbf{1},\mathbf{1}) + (1-p)^2 K(\mathbf{0},\mathbf{0}) + 2p(1-p)K(\mathbf{1},\mathbf{0})$$
$$\quad + q^2 K(\mathbf{1},\mathbf{1}) + (1-q)^2 K(\mathbf{0},\mathbf{0}) + 2q(1-q)K(\mathbf{1},\mathbf{0})$$
$$\quad - 2\left(pqK(\mathbf{1},\mathbf{1}) + (1-p)(1-q)K(\mathbf{0},\mathbf{0}) + 2(p(1-q) + q(1-p))K(\mathbf{1},\mathbf{0})\right)$$
$$= (K(\mathbf{1},\mathbf{1}) + K(\mathbf{0},\mathbf{0}) - 2K(\mathbf{1},\mathbf{0}))(p-q)^2 \tag{29}$$

Moreover, for any $\mathbb{P}, \mathbb{Q} \in \mathcal{M}^*$, we also have that

$$\mathrm{KL}\left(\mathbb{Q} \parallel \mathbb{P}\right) = q \log\left(\frac{q}{p}\right) - (1-q) \log\left(\frac{1-p}{1-q}\right)$$

$$\leq (q-p)\left[\frac{q}{p} - \frac{1-q}{2-(p+q)}\right] \tag{30a}$$

$$\leq \frac{17}{6}(q-p) \tag{30b}$$

$$\leq \frac{34}{3}(q-p)^2, \tag{30c}$$

where (30a) holds due to the bound $\frac{x-1}{x+1} \leq \log(x) \leq x - 1$ for $x \geq 1$, (30b) holds because $p, q \in [1/4, 3/4]$, and (30c) holds due to the bound $x \leq 4x^2$ for $x \geq 1/4$. Combining (29) and (30c), and additionally making use of the shift invariance of $K$, we obtain that

$$\mathrm{KL}\left(\mathbb{Q} \parallel \mathbb{P}\right) \leq \frac{17}{3}\left(K\left(\mathbf{0}, \mathbf{0}\right) - K\left(\mathbf{1}, \mathbf{0}\right)\right)^{-1} \mathrm{MMD}_K^2\left[\mathbb{P}, \mathbb{Q}\right]. \tag{31}$$

Therefore, putting $C_K = (1/2)(3/17)\left(K\left(\mathbf{0}, \mathbf{0}\right) - K\left(\mathbf{1}, \mathbf{0}\right)\right)$ for the constant in (12), using (31) along with the fact that $\mathcal{M}^* \subset \mathcal{M}_1^+\left(\mathbb{R}^d\right)$, we have that

$$\text{L.H.S. of (12)} \geq \inf_{N: \mathbb{P}_\infty(N \leq \infty) \leq \alpha} \sup_{\substack{\eta > 1 \\ \mathbb{P}, \mathbb{Q} \in \mathcal{M}^*}} \mathbb{P}\left(N \geq \eta + \frac{(1/2)\log\left(1/\alpha\right)}{\mathrm{KL}\left(\mathbb{Q} \parallel \mathbb{P}\right)}\right). \tag{32}$$

Consequently the theorem is proved if we can find absolute constants $\alpha_0, \beta_0 \in (0,1)$ and pre- and post-change distributions $\mathbb{P}, \mathbb{Q} \in \mathcal{M}^*$ such that for all $\alpha \leq \alpha_0$ it holds that

$$\inf_{N: \mathbb{P}_\infty(N \leq \infty) \leq \alpha} \sup_{\eta > 1} \mathbb{P}\left(N \geq \eta + \frac{(1/2)\log\left(1/\alpha\right)}{\mathrm{KL}\left(\mathbb{Q} \parallel \mathbb{P}\right)}\right) \geq \beta_0. \tag{33}$$

To show (33), one can use a change of measure argument originally due to Lai [30]. In fact one can directly use the version of Lai's argument adapted to finite sample analysis by Yu et al. [64, Proposition 4.1]. For clarity of exposition, we repeat the argument below. The following holds for arbitrary $\mathbb{P}, \mathbb{Q} \in \mathcal{M}^*$. For each $n \in \mathbb{N}$ let $\mathcal{F}_n$ be the $\sigma$-field generated by $\{X_i\}_{i=1}^n$ and let $\mathbb{P}^{\otimes n}$ be the restriction of the joint law to $\mathcal{F}_n$. We can write

$$\frac{\mathrm{d}\mathbb{P}_\eta^{\otimes n}}{\mathrm{d}\mathbb{P}_\infty^{\otimes n}} = \exp\left(\sum_{i=1}^n Z_i\right), \qquad \text{for } n > \eta$$

where, as in the main text, the subscripts indicate the time at which the change occurs. For a chosen $\alpha \in (0,1)$ and an arbitrary stopping time satisfying $\mathbb{P}_\infty(N < \infty) \leq \alpha$ introduce the events

$$\mathcal{E}_1 = \left\{\eta \leq N \leq \eta + \frac{(1/2)\log\left(1/\alpha\right)}{\mathrm{KL}\left(\mathbb{Q} \parallel \mathbb{P}\right)}, \sum_{i=\eta+1}^N Z_i \leq (3/4)\log\left(1/\alpha\right)\right\}$$

$$\mathcal{E}_2 = \left\{\eta \leq N \leq \eta + \frac{(1/2)\log\left(1/\alpha\right)}{\mathrm{KL}\left(\mathbb{Q} \parallel \mathbb{P}\right)}, \sum_{i=\eta+1}^N Z_i > (3/4)\log\left(1/\alpha\right)\right\}.$$

For the first event we have that

$$\mathbb{P}_\eta\left(\mathcal{E}_1\right) = \int_{\mathcal{E}_1} \exp\left(\sum_{i=1}^N Z_i\right) \mathrm{d}\mathbb{P}_\eta \leq \exp\left((3/4)\log\left(1/\alpha\right)\right) \mathbb{P}_\infty\left(\mathcal{E}_1\right) \leq \alpha^{1/4} \tag{34}$$

where the first inequality is due to the definition of $\mathcal{E}_1$ and the second inequality holds because the probability of $N$ being finite when no change occurs is bounded from above by $\alpha$. For the second

event we have that

$$
\mathbb{P}_\eta \left( \mathcal{E}_2 \right) \leq \mathbb{P}_\eta \left( \bigcup_{1=t}^{(1/2)\log(1/\alpha)(\mathrm{KL}(\mathbb{Q}\|\mathbb{P}))^{-1}-1} \sum_{i=\eta+1}^{\eta+t} Z_i > (3/4)\log\left(1/\alpha\right) \right)
$$

$$
= \mathbb{P}_\eta \left( \bigcup_{1=t}^{(1/2)\log(1/\alpha)(\mathrm{KL}(\mathbb{Q}\|\mathbb{P}))^{-1}-1} \sum_{i=\eta+1}^{\eta+t} \left( Z_i - \mathrm{KL}\left(\mathbb{Q}\parallel\mathbb{P}\right)\right) > (1/4)\log\left(1/\alpha\right) \right) \tag{35a}
$$

$$
\leq \frac{(1/2)\log(1/\alpha)}{\mathrm{KL}\left(\mathbb{Q}\parallel\mathbb{P}\right)} \exp\left(-\log(1/\alpha)\right) \tag{35b}
$$

$$
\leq \alpha^{1/4} \tag{35c}
$$

where in particular (35a) holds by subtracting $t \times \mathrm{KL}\left(\mathbb{Q}\parallel\mathbb{P}\right)$ from both sides of the inequality and using the fact that for every $t$ in the union it holds that $t \times \mathrm{KL}\left(\mathbb{Q}\parallel\mathbb{P}\right) < (1/2)\log(1/\alpha)$, (35b) holds due to a union bound argument followed by an application of Hoeffding's inequality, and (35c) holds for all $\alpha \leq \alpha_0$ where

$$
\alpha_0 = \sup\left\{ \alpha \in (0,1) \mid (1/2)\log(1/\alpha)\alpha^{3/4} \leq \inf_{\mathbb{P},\mathbb{Q}\in\mathcal{M}^*} \mathrm{KL}\left(\mathbb{Q}\parallel\mathbb{P}\right) \text{ and } 2\alpha^{1/4} < 1 \right\}.
$$

Since the above arguments do not depend on the stopping time $N$ or the change point location $\eta$, the bounds (34) and (35c) together imply that R.H.S. of (32) $\geq 1 - 2\alpha_0^{1/4}$, which proves the desired result. $\qquad\square$

### A.4.5 Proof of Lemma 1

*Proof.* It suffices to show that for any $\epsilon > 0$ it holds that

$$
\mathbb{P} \otimes \Lambda \left( \left\| \frac{1}{n} \sum_{i=1}^n \hat{z}_K(X_i) - \mathbb{E}_{|\omega}\hat{z}_K(X) \right\| > \epsilon \right) \leq 2\exp\left( -\frac{1}{2}\frac{n\varepsilon^2}{\tilde{\sigma}^2 + 2\varepsilon} \right), \tag{36}
$$

which implies the stated result by using $\pm\mathbb{E}_{|\omega}\hat{z}_K(X)$, the triangle inequality, and a union bound. Here and in the following $\mathbb{E}_{|\omega}$ (resp. $\mathbb{P}_{|\omega}$) refers to the expectation (resp. probability) with $\omega$ fixed. To prove (36), we fix $\omega = (\omega_1, \ldots, \omega_r)$ and let $\eta_i = \hat{z}_K(X_i) - \mathbb{E}_{|\omega}\hat{z}_K(X)$. Then, all $\eta_i$-s are zero mean w.r.t. $\mathbb{P}$ and i.i.d. Further, we have for any $p \geq 2$ that

$$
\mathbb{E}_{|\omega}\|\eta_1\|^p = \mathbb{E}_{|\omega}\|\eta_1\|^{p-2}\|\eta_1\|^2 \leq \sup\|\eta_1\|^{p-2}\mathbb{E}_{|\omega}\|\eta_1\|^2 \leq 2^{p-2}\mathbb{E}_{|\omega}\|\eta_1\|^2,
$$

where the last inequality follows by using the triangle inequality and $\langle\hat{z}_K(X_1), \hat{z}_K(X_1)\rangle = 1$ to obtain

$$
\sup\|\eta_1\|^{p-2} \leq 2^{p-2}\sup\|\hat{z}_K(X_1)\|^{p-2} = 2^{p-2}.
$$

Hence, we have the bound

$$
\mathbb{E}_{|\omega}\|\eta_1\|^p \leq 2^{p-2}\mathbb{E}_{|\omega}\|\eta_1\|^2 \leq \frac{1}{2}p!2^{p-2}\mathbb{E}_{|\omega}\|\eta_1\|^2
$$

and the $\eta_i$-s satisfy the Bernstein condition with $B^2 = \mathbb{E}_{|\omega}\|\eta_1\|^2$ and $H = 2$. The application of Theorem 8 yields that for any $\varepsilon > 0$

$$
\mathbb{P}_{|\omega} \left( \left\| \frac{1}{n} \sum_{i=1}^n \eta_i \right\| > \epsilon \right) \leq 2\exp\left( -\frac{1}{2}\frac{n^2\varepsilon^2}{B^2 + \varepsilon nH} \right) \leq 2\exp\left( -\frac{1}{2}\frac{n\varepsilon^2}{B^2 + \varepsilon H} \right).
$$

To lift the conditioning, we observe that

$$
\mathbb{P} \otimes \Lambda \left( \left\| \frac{1}{n} \sum_{i=1}^n \eta_i \right\| > \epsilon \right) = \mathbb{E}\mathbb{P}_{|\omega} \left( \left\| \frac{1}{n} \sum_{i=1}^n \eta_i \right\| > \epsilon \right) \leq 2\mathbb{E}\exp\left( -\frac{1}{2}\frac{n\varepsilon^2}{B^2 + \varepsilon H} \right)
$$

$$
\leq 2\exp\left( -\frac{1}{2}\frac{n\varepsilon^2}{\mathbb{E}B^2 + \varepsilon H} \right),
$$

where the last inequality holds by the concavity of $\exp(-1/x)$ for $x > 0$ and Jensen's inequality. Finally, we conclude the proof by using that

$$\mathbb{E}B^2 = \mathbb{E}\Big(\langle \hat{z}_K(X), \hat{z}_K(X)\rangle + \langle \mathbb{E}_\mathbb{P}\hat{z}_K(X), \mathbb{E}_\mathbb{P}\hat{z}_K(Y)\rangle - 2\langle \hat{z}_K(X), \mathbb{E}_\mathbb{P}\hat{z}_K(Y)\rangle\Big)$$
$$= \mathbb{E}K(X, X) - \mathbb{E}K(X, Y) = \tilde{\sigma}^2,$$

where the boundedness of all terms allows exchanging the expectations by the Fubini-Tonelli theorem.

$\square$

### A.4.6 Proof of Theorem 5

*Proof.* We first show that with probability at least $1 - \alpha$ no local tests conducted on null intervals will cause the algorithm to wrongly declare a change. For each $n \leq T$ define $k_n = \max\{k \leq M+1 \mid \eta_k \leq n\}$. With $\pi_n$ defined as in (17), introduce the event

$$A_1 = \left\{\cap_{n=2}^{T} \cap_{j=1}^{\lfloor \log_2(n-\eta_{k_n}+1)\rfloor-1} \hat{M}_{n,j} \leq \pi_n\right\}. \tag{37}$$

We therefore have that

$$\mathbb{P}(A_1^c) \leq \sum_{l=1}^{T} 2^l \max_{2^l \leq n \leq 2^{l+1}} \sum_{j=0}^{\lfloor \log_2(n-\eta_{k_n}+1)\rfloor-1} \mathbb{P}\left(\hat{M}_{n,j} > \pi_n\right) \tag{38a}$$

$$\leq \sum_{l=1}^{T} l2^l e^{-(\pi_{2^l})^2/2} \tag{38b}$$

$$= \frac{\alpha \log_2(T)}{1 + \log_2(T)} < \alpha,$$

where (38a) follows by arguments similar to (18a) and (38b) holds due to Lemma 5. Therefore, if $M = 0$ we are done. We now tackle the case $M > 0$. We will show that when $M > 0$ with probability at least $1 - 2\alpha$ all true change points are detected and localized at the stated rate. We first make the constants in Theorem 5 explicit:

$$C_1 = 2 \times C_3$$
$$C_2 = 4$$
$$C_3 = \left(2 \times \left(\sqrt{2} + \sqrt{4} + \sqrt{6}\right)\right)^2.$$

Introduce also the event

$$A_2 = \left\{\sup_{\mathbf{x},\mathbf{y}\in\mathcal{X}} \left|\hat{K}(\mathbf{x},\mathbf{y}) - K(\mathbf{x},\mathbf{y})\right| \leq \frac{h(d,|\mathcal{X}|,\sigma) + \sqrt{2\log(2/\alpha)}}{\sqrt{r}}\right\}, \tag{39}$$

on which the random Fourier feature kernel $\hat{K}$ is close to $K$ in sup norm sense. Note that by Theorem 7 the event (39) holds with probability at least $1 - \alpha/2$. Introduce also the sequence of events

$$B_k = \left\{\eta_k < \hat{\eta}_k \leq \eta_k + C_3 \frac{\log(T/\alpha)}{(\mathrm{MMD}_K[\mathbb{Q}_k, \mathbb{Q}_{k+1}])^2}\right\}, \quad \text{for} \quad k = 1,\ldots M,$$

such that on the event $B_k$ the $k$-th change point is detected and localized at the required rate. If $\hat{M} < M$ we employ the convention $\hat{\eta}_k = T$ for all $k > \hat{M}$. We now show that on (37) and (39) the event $B_1$ holds with probability at least $T/(2\alpha)$. Introduce the quantity

$$n_1^* = \min\left\{n < \eta_2 \mid n - \eta_1 = 2^{j_1^*} \text{ for } j_1^* \in \mathbb{N}, \sqrt{2^{j_1^*}}\mathrm{MMD}_K[\mathbb{Q}_1, \mathbb{Q}_2] > \sqrt{C_3 \log(2T/\alpha)}\right\}$$

To ease the notation put $I_1^* = 1 : \eta_1$, $J_1^* = (\eta_1 + 1) : n_1^*$, and $V_1^* = \frac{2^{j_1^*}(n_1^* - 2^{j_1^*})}{n_1^*}$. Observe that

$$
\sqrt{V_1^*}\mathrm{MMD}_{\hat{K}}\left[X_{I_1^*}, X_{J_1^*}\right]
$$

$$
\geq \sqrt{V_1^*}\mathrm{MMD}_K\left[\mathbb{Q}_1, \mathbb{Q}_2\right] + \sqrt{V_1^*}\left(\mathrm{MMD}_K\left[X_{I_1^*}, X_{J_1^*}\right] - \mathbb{E}\left[\mathrm{MMD}_K\left[X_{I_1^*}, X_{J_1^*}\right]\right]\right) \quad (40a)
$$

$$
- 2\sqrt{V_1^*}\sqrt{\sup_{\boldsymbol{x},\boldsymbol{y}}\left|\hat{K}(\boldsymbol{x},\boldsymbol{y}) - K(\boldsymbol{x},\boldsymbol{y})\right|}
$$

$$
- \sqrt{V_1^*}\left|\mathbb{E}\left[\mathrm{MMD}_K\left[X_{I_1^*}, X_{J_1^*}\right]\right] - \mathrm{MMD}_K\left[\mathbb{Q}_1, \mathbb{Q}_2\right]\right|
$$

$$
\geq \frac{1}{2}\sqrt{V_1^*}\mathrm{MMD}_K\left[\mathbb{Q}_1, \mathbb{Q}_2\right] + \sqrt{V_1^*}\left(\mathrm{MMD}_K\left[X_{I_1^*}, X_{J_1^*}\right] - \mathbb{E}\left[\mathrm{MMD}_K\left[X_{I_1^*}, X_{J_1^*}\right]\right]\right) - \frac{4}{\sqrt{2}}
$$

$$(40b)$$

Where (40a) holds by the triangle inequality and arguments similar to (21), and (40b) holds by arguments similar to (22) together with the fact that on (39) we will have that

$$
\sup_{\boldsymbol{x},\boldsymbol{y}}\left|\hat{K}(\boldsymbol{x},\boldsymbol{y}) - K(\boldsymbol{x},\boldsymbol{y})\right| \leq \frac{1}{4}\min_{k=1,\dots,M}\left(\mathrm{MMD}_K\left[\mathbb{Q}_k, \mathbb{Q}_{k+1}\right]\right)^2 \leq \frac{1}{4}\left(\mathrm{MMD}_K\left[\mathbb{Q}_1, \mathbb{Q}_2\right]\right)^2.
$$

Using the above we therefore have that

$$
\mathbb{P}\left(B_1^c\right) \leq \mathbb{P}\left(\sqrt{V_1^*}\mathrm{MMD}_{\hat{K}}\left[X_{I_1^*}, X_{J_1^*}\right] \leq \lambda_T\right)
$$

$$
\leq \mathbb{P}\left(\sqrt{V_1^*}\left(\mathbb{E}\left[\mathrm{MMD}_{\hat{K}}\left[X_{I_1^*}, X_{J_1^*}\right]\right] - \mathrm{MMD}_{\hat{K}}\left[X_{I_1^*}, X_{J_1^*}\right]\right)\right.
$$

$$
\left. > \frac{1}{2}\sqrt{V_1^*}\mathrm{MMD}_K\left[\mathbb{Q}_1, \mathbb{Q}_2\right] - \left(\sqrt{4} + \sqrt{6}\right)\sqrt{\log(2T/\alpha)}\right) \leq \frac{\alpha}{2T},
$$

where the final inequality follows from the definition of $C_3$, the bounded difference property of $\mathrm{MMD}_K$, and Theorem 6. By the definition we must have that $2^{j_1^*} \leq \delta_1/2$, therefore on the event $A_1$ the first change point is detected and the algorithm starts at most mid way between $\eta_1$ and $\eta_2$. Identical arguments to those above therefore give that, on the events $A_1$, $A_2$, and $B_1$ the event $B_2$ holds with probability at least $1 - \alpha/(2T)$ and having detected the second change point the algorithm re-starts at most mid way between $\eta_2$ and $\eta_3$. By induction and a union bound argument, on the events (37) and (39) the events $B_1, \dots B_M$ hold with probability at least $1 - M\alpha/(2T) \geq 1 - \alpha/2$. Since (37) and (39) jointly hold with probability $1 - 3\alpha/2$ we are done. □

## A.5 Auxiliary results

In this section, we collect a few auxiliary results. Besides establishing useful bounds on real numbers in Lemma 2 and Lemma 3, we show that the bounded differences property of RFF-MMD (Lemma 4) leads to its exponential concentration (Lemma 5). The latter is one of the key ingredients for deriving our threshold sequences elaborated Section 4.3.

**Lemma 2.** *For any $x, y > 0$ it holds that*

$$
\frac{1}{2}\min\left(x, y\right) \leq \frac{xy}{x+y} \leq \min\left(x, y\right),
$$

*and, moreover, both inequalities are tight.*

*Proof.* We first note that

$$
\frac{xy}{x+y} = \frac{\min\left(x, y\right)\max\left(x, y\right)}{\min\left(x, y\right) + \max\left(x, y\right)} = \min\left(x, y\right)\left(1 + \frac{\min\left(x, y\right)}{\max\left(x, y\right)}\right)^{-1}.
$$

For the lower bound we use the fact that $1 + \frac{\min(x,y)}{\max(x,y)} \leq 2$, where equality holds when $x = y$. For the upper bound, we use that $1 + \frac{\min(x,y)}{\max(x,y)} \geq 1$, where equality holds in the limit when, for instance, $x$ is fixed and $y \to +\infty$. □

**Lemma 3.** *For $x, y > 0$ it holds that $\left|\sqrt{x} - \sqrt{y}\right| \leq \sqrt{|x - y|}$.*

*Proof.* When $x = y$ the statement is trivially true. When $x \neq y$ it holds that

$$\left| \sqrt{x} - \sqrt{y} \right| = \frac{|x - y|}{\sqrt{x} + \sqrt{y}} \leq \frac{|x - y|}{\left| \sqrt{x} - \sqrt{y} \right|} \Rightarrow \left| \sqrt{x} - \sqrt{y} \right| \leq \sqrt{|x - y|}.$$

$\square$

**Lemma 4.** *The RFF-MMD as defined in (7) between two empirical measures composed respectively of $n$ and $m$ sample points is a function mapping from $(\mathbb{R}^d)^{m+n} \to \mathbb{R}$. This function has the bounded differences property with constants*

$$c_i = \begin{cases} 2/n & \text{if } i = 1, \ldots, n, \\ 2/m & \text{if } i = n + 1, \ldots, n + m. \end{cases} \tag{41}$$

*Proof.* Recall that if $K : \mathbb{R}^d \times \mathbb{R}^d \to \mathbb{R}$ is the reproducing kernel for some RKHS $\mathcal{H}_K$, for any $\mathbb{P}, \mathbb{Q} \in \mathcal{M}_1^+$, one has that

$$\text{MMD}_K\left[ \mathbb{P}, \mathbb{Q} \right] = \sup_{f \in \mathcal{H}_K : \|f\|_{\mathcal{H}_K} \leq 1} \left( \mathbb{E}_{X \sim \mathbb{P}} \left[ f(X) \right] - \mathbb{E}_{Y \sim \mathbb{Q}} \left[ f(Y) \right] \right)$$

Note that $\hat{K} : \mathbb{R}^d \times \mathbb{R}^d \to \mathbb{R}$ as defined in (6) is the reproducing kernel for an RKHS $\mathcal{H}_{\hat{K}}$ whose elements are vectors in $\mathbb{R}^{2r}$. Introduce the set

$$\hat{\mathcal{G}} = \{ f \in \mathcal{H}_{\hat{K}} \mid \|f\|_{\mathcal{H}_{\hat{K}}} \leq 1 \}.$$

Let $\text{MMD}_{\hat{K}}(\mathbf{x}_{1:n}, \mathbf{y}_{1:m})(\tilde{x}_{i'})$ stand for (7) with inputs $\{\mathbf{x}_1, \ldots, \mathbf{x}_n\}$ and $\{\mathbf{y}_1, \ldots, \mathbf{y}_m\}$ with the $i'$-th $\mathbf{x}$ replaced by $\tilde{\mathbf{x}}_{i'}$. We therefore have that

$$\sup_{\substack{\mathbf{x}_1, \ldots, \mathbf{x}_n \\ \mathbf{y}_1, \ldots, \mathbf{y}_m \\ \tilde{\mathbf{x}}_{i'}}} \left| \text{MMD}_{\hat{K}}\left[ \mathbf{x}_{1:n}, \mathbf{y}_{1:m} \right] - \text{MMD}_{\hat{K}}\left[ \mathbf{x}_{1:n}, \mathbf{y}_{1:m} \right](\tilde{x}_{i'}) \right|$$

$$= \sup_{\substack{\mathbf{x}_1, \ldots, \mathbf{x}_n \\ \mathbf{y}_1, \ldots, \mathbf{y}_m \\ \tilde{\mathbf{x}}_{i'}}} \left| \sup_{f \in \hat{\mathcal{G}}} \left( \frac{1}{n} \sum_{i=1}^{n} f(\mathbf{x}_i) - \frac{1}{m} \sum_{j=1}^{m} f(\mathbf{y}_i) \right) \right.$$

$$\left. - \sup_{f \in \hat{\mathcal{G}}} \left( \frac{1}{n} \sum_{i=1}^{n} f(\mathbf{x}_i) - \frac{1}{n} \left[ f(\mathbf{x}_{i'}) - f(\tilde{\mathbf{x}}_{i'}) \right] - \frac{1}{m} \sum_{j=1}^{m} f(\mathbf{y}_i) \right) \right|$$

$$\leq \sup_{\substack{\mathbf{x}, \tilde{\mathbf{x}} \\ f \in \hat{\mathcal{G}}}} \frac{1}{n} \left| f(\mathbf{x}) - f(\tilde{\mathbf{x}}) \right| = \sup_{\mathbf{x}, \tilde{\mathbf{x}}} \frac{1}{n} \left\| \hat{K}(\cdot, \mathbf{x}) - \hat{K}(\cdot, \tilde{\mathbf{x}}) \right\|_{\hat{\mathcal{H}}}$$

$$\leq \sup_{\mathbf{x}, \tilde{\mathbf{x}}} \frac{1}{n} \left( \left\| \hat{K}(\cdot, \mathbf{x}) \right\|_{\hat{\mathcal{H}}} + \left\| \hat{K}(\cdot, \tilde{\mathbf{x}}) \right\|_{\hat{\mathcal{H}}} \right) = \frac{2}{n}$$

where we used the reverse triangle inequality, the reproducing property, CBS for obtaining the supremum over a unit ball and that, for any $\mathbf{x} \in \mathbb{R}^d$ one has that

$$\left\langle \hat{K}(\cdot, \mathbf{x}), \hat{K}(\cdot, \mathbf{x}) \right\rangle_{\mathcal{H}_{\hat{K}}}^2 = \cos(0) = 1.$$

The same calculations can be applied to $\text{MMD}_{\hat{K}}(\mathbf{x}_{1:n}, \mathbf{y}_{1:m})(\tilde{\mathbf{y}}_{j'})$. This proves the desired result.

$\square$

**Lemma 5.** *Given two independent samples $\{X_1, \ldots, X_n\}$ and $\{Y_1, \ldots, Y_m\}$, each with mutually independent entries drawn from some $\mathbb{P} \in \mathcal{M}_1^+$, for any $\varepsilon > 0$, it holds that*

$$\mathbb{P} \left( \sqrt{\frac{nm}{n+m}} \, \text{MMD}_{\hat{K}}[X_{1:n}, Y_{1:m}] > \sqrt{2} + \varepsilon \right) \leq e^{-\varepsilon^2/2}.$$

*Proof.* It is an immediate consequence of Lemma 4 and Theorem 6 that for any $\varepsilon' > 0$

$$\mathbb{P}\left(\text{MMD}_{\hat{K}}[X_{1:n}, Y_{1:m}] - \mathbb{E}\left[\text{MMD}_{\hat{K}}[X_{1:n}, Y_{1:m}] \mid \omega_1, \ldots, \omega_r\right] > \varepsilon' \mid \omega_1, \ldots, \omega_r\right)$$

$$\leq \exp\left(-\frac{\varepsilon'^2}{2}\frac{nm}{n+m}\right). \tag{42}$$

Moreover, arguing as in the last step of the proof of Proposition 4 in [23] gives

$$\mathbb{E}\left[\text{MMD}_{\hat{K}}[X_{1:n}, Y_{1:m}] \mid \omega_1, \ldots, \omega_r\right]$$

$$= \mathbb{E}\left[\left(\frac{1}{n^2}\sum_{i=1}^{n}\sum_{j=1}^{n}\hat{K}(X_i, X_j) + \frac{1}{m^2}\sum_{i=1}^{m}\sum_{j=1}^{m}\hat{K}(Y_i, Y_j)\right.\right.$$

$$\left.\left.-\frac{2}{nm}\sum_{i=1}^{n}\sum_{j=1}^{m}\hat{K}(X_i, Y_j)\right)^{\frac{1}{2}} \mid \omega_1, \ldots, \omega_r\right]$$

$$\leq \left(\left\{\frac{1}{n} + \frac{1}{m}\right\}\underbrace{\mathbb{E}\left[\hat{K}(X, X) \mid \omega_1, \ldots, \omega_r\right]}_{=1}\right.$$

$$\left.+ \underbrace{\left\{\frac{n-1}{n} + \frac{m-1}{m} - 2\right\}}_{=-\left(\frac{1}{n}+\frac{1}{m}\right)}\mathbb{E}\left[\hat{K}(X, Y) \mid \omega_1, \ldots, \omega_r\right]\right)^{\frac{1}{2}}$$

$$= \left(\left\{\frac{1}{n} + \frac{1}{m}\right\}\left(1 - \underbrace{\mathbb{E}\left[\hat{K}(X, Y) \mid \omega_1, \ldots, \omega_r\right]}_{\geq -1}\right)\right)^{\frac{1}{2}}$$

$$\leq \sqrt{\frac{2(m+n)}{mn}}, \tag{43}$$

where the first inequality follows from Jensen's inequality. Consequently, setting $\varepsilon' = \varepsilon\sqrt{\frac{n+m}{nm}}$, plugging (43) into (42), and integrating over the $\omega$-s with respect to the product measure $\Lambda^{\otimes r} := \Lambda \otimes \cdots \otimes \Lambda$ yields the desired result. $\square$

## A.6 External statements

In this section, we collect the external statements that we use, to ensure self-completeness. Theorem 6 recalls McDiarmid's inequality from Boucheron et al. [5, Section 6.1], which is also known as bounded differences inequality [38]. Theorem 7 is about the concentration of random Fourier features and part of the proof of Sriperumbudur and Szabó [54, Theorem 1]. We recall the concentration result Yurinsky [65, Theorem 3.3.4] on random variables taking values in a separable Hilbert space in Theorem 8.

**Theorem 6** (Bounded differences inequality). *Let $\mathcal{X}$ be a measurable space. A function $f : \mathcal{X}^n \to \mathbb{R}$ has the bounded difference property for some constants $c_1, \ldots, c_n$ if, for each $i = 1, \ldots, n$,*

$$\sup_{\substack{x_1, \ldots, x_n \in \mathcal{X} \\ x_i' \in \mathcal{X}}} |f(x_1, \ldots, x_{i-1}, x_i, x_{i+1}, x_n) - f(x_1, \ldots, x_{i-1}, x_i', x_{i+1}, \ldots x_n)| \leq c_i. \tag{44}$$

*Then, if $X_1, \ldots, X_n$ is a sequence of independently distributed random variables and (44) holds, putting $Z = f(X_1, \ldots, X_n)$ and $\nu = \frac{1}{4}\sum_{i=1}^{n}c_i^2$ for any $t > 0$, it holds that*

$$\mathbb{P}(Z - \mathbb{E}(Z) > t) \leq e^{-t^2/(2\nu)}.$$

**Theorem 7** (RFF exponential concentration). *Let $\hat{K}$ be defined as in (6). Let $\mathcal{X}$ be a proper subset of $\mathbb{R}^d$ and denote by $|\mathcal{X}|$ its Lebesgue measure. For any $t > 0$, it holds that*

$$\mathbb{P}\left(\sup_{\mathbf{x}, \mathbf{y} \in \mathcal{X}}\left|\hat{K}(\mathbf{x}, \mathbf{y}) - K(\mathbf{x}, \mathbf{y})\right| > \frac{h(d, |\mathcal{X}|, \sigma) + t}{\sqrt{r}}\right) \leq e^{-t^2/2}$$

*where $\sigma^2 = \int_{\mathbb{R}^d} \|\omega\|_2^2 \, \mathrm{d}\Lambda(\omega)$ and*

$$h(d, |\mathcal{X}|, \sigma) = 23\sqrt{2d \log(2|\mathcal{X}| + 1)} + 32\sqrt{2d \log(\sigma + 1)} + 16\sqrt{2d \left[\log(2|\mathcal{X}| + 1)\right]^{-1}}. \quad (45)$$

**Theorem 8** (Hilbert space Bernstein inequality). *Let $X_1, \ldots, X_n$ be a sequence of zero mean independent random variables taking values in a real and separable Hilbert space $\mathcal{X}$ with inner product $\langle \cdot, \cdot \rangle$ and norm $\|\cdot\| = \sqrt{\langle \cdot, \cdot \rangle}$. Write $S_n^* = \sup_{m \leq n} \|X_1 + \cdots + X_m\|$. If the random variables satisfy the moment condition*

$$\mathbb{E} \|X\|^k \leq \frac{1}{2} k! B^2 H^{k-2}, \qquad \text{for } k \geq 2, i = 1, \ldots, n$$

*for some constants $B > 0$ and $H > 0$, then for any $x > 0$ it holds that*

$$\mathbb{P}(S_n^* > xB) \leq 2 \exp\left(-\frac{1}{2}x^2 \left[1 + \frac{xH}{B}\right]^{-1}\right).$$

