# OpenReview forum: "Optimal Online Change Detection via Random Fourier Features"
_NeurIPS.cc/2025/Conference — NeurIPS 2025 poster_

### Official Review · Reviewer_4v7u · 2025-06-25

**Clarity:** 3
**Significance:** 2
**Originality:** 3
**Rating:** 5
**Confidence:** 4

**Summary:**

The paper addresses a problem of online non-parametric change point detection. More precisely, the authors advance existing change point detection kernel-based methods and propose to approximate Maximum Mean Discrepancy using random Fourier features. Such an approach results in a faster procedure compared to established Scan B-statistics and does not require any historical data or a specification of window parameters. Theoretical results guarantee the control of the average running length or, alternatively, false alarm probability, bounding of detection delay, and its optimality in the minimax perspective. Experimental results also confirm the superiority of the method compared to several baselines.

**Questions:**

**Main questions:**

(1) Do the authors consider other real-world datasets common for change point detection problems? What was the model's performance?

(2) In the work, the authors use a Gaussian kernel for all experiments. Do the conclusions the same for other kernels? How do empirical results change when the used kernel does not satisfy Assumption 1?

Just an additional comment. According to line 94, the authors consider independent random variables. It is another limitation of the work not mentioned in the corresponding section. However, I understand that taking into account the dependency of the data for change point detection is still a challenging task.

**Minor comments:**

(1) Footnote 1 should be placed near Table 1. Now, it is located on the next pages, thereby complicating understanding;
(2) Line 216,"In the following..." missing comma.

**Ethical Concerns:**

["NO or VERY MINOR ethics concerns only"]

**Final Justification:**

In general, I have no sensitive concerns about the papers before rebuttal.
I will recommend accepting the paper.

**Limitations:**

Yes

**Quality:**

3

**Strengths And Weaknesses:**

**Strengths**

The paper represents strong and consistent scientific work. The presentation is concise, the text is well-written, and the mathematician language is rigorous and accurate with all necessary articulation of corresponding attributes. At the same time, main concepts are explained in natural collocations, thereby enhancing the overall comprehension of the text. Although this research is closely connected with previous works (e.g., Scan B-statistic, M-statistic, KL-CPD), the proposed method overcomes several challenges through the framework of random Fourier features. Thus, my overall impression is good.

**Weaknesses**

My main concern is devoted to the experimental part. The work considers two types of data: synthetic sets and MNIST sequences. Actually, neither of them corresponds to real-world tasks. The latent manifold of MNIST images is relatively simple and can be separated efficiently, even with linear methods. Therefore, the performance of the method on real-world data (at least to some extent from the real world) remains underexplored. One of the most interesting scenarios is the detection of changes in video data mentioned in line 21. Therefore, I recommend considering such data, but different typical low-dimensional sequences for CPD (e.g., Human Activity Recognition, EEG signals, financial data) are also worthy of consideration. Moreover, although I understand how the authors choose their baselines for comparison, I would rather want to see a wider range of competitors to make sure the novel approach is a new "state-of-the-art".

---

> ### Author Rebuttal · Authors · 2025-07-31
>
> We thank the reviewer for the time and effort invested and for the valuable feedback. Below, we answer all questions in detail.
>
> __Other real world data set: MazurkaBL.__ To further validate our algorithm's performance, we additionally ran the proposed method on the "M17-4" sample of the MazurkaBL [Kosta et al., 2018] data set with the goal of detecting changes in the so-called "sones" (see their "scripts/ismir\_tutorial.ipynb"); as in the article, we use the Gaussian kernel with the median heuristic. For obtaining the thresholds using Monte Carlo iterations, we slice the sone data along each change point and compute the test statistics obtained by the proposed method individually on each one. We then select the $1-0.1$ quantile across all test statistics so obtained as our threshold.
>
> To then search for change points, we again slice the sone data, but in this case such that each slice contains precisely one change point, which yields a total of $25$ samples. We process each one of these with our proposed method and consider the first time the test statistic exceeds the threshold as detected change. In total, our proposed method flags $10$ change points too early, and, on the remaining $15$ has an average detection delay of $93.4$, with a median detection delay of $74.0$.
>
> We also refer to our additional experiments on the popular HASC data set in the response to reviewer ``ArQr''.
>
>
> __Non-Gaussian kernels.__ Our theoretical results hold for any continuous bounded characteristic translation-invariant kernel, that is, any kernel that satisfies Assumption 1. Kernel functions that do not satisfy Assumption 1 (for example the unbounded exponential kernel; Chamakh and Szabó 2021) do not permit the Bochner representation (7) and hence can not be approximated by random Fourier features. In this case our method is not applicable.
> However, we emphasize that the class of kernels satisfying Assumption 1 is large and, in particular, encompasses most kernels typically employed on Euclidean spaces. For example, the considered class includes Gaussian kernels, mixtures of Gaussian kernels, inverse multi-quadratic kernels, Matérn kernels, or Laplace kernels.
>
> While, as indicated, all our results hold for these kernels, we recall from Section 7 that the particular choice of kernel influences the online analogue of testing power. Hence, in practice, one chooses a kernel function that captures the characteristics of the data. The Gaussian kernel with the median heuristic, used in our experiments, is a sensible default.
>
>
> __Non-i.i.d. random sequences.__ We agree that non-i.i.d.\ sequences occur in practice and that considering only independently distributed data can be seen as a limitation. Intuitively, we believe that analogues to Theorems 1--3 can be obtained if one were to replace the independence assumption with a suitable notion of weak dependence such as strong mixing g [Bradley, 2005] or functional dependence [Wu, 2005]. However, we do not have a clear idea of how minimax optimality results in this setting (i.e., an analogue to Theorem 4) would look like. Hence, extending our results to non-i.i.d. sequences is interesting future work.
>
> We hope these answers fully settle all open questions.
>
> Linda Chamakh and Zoltán Szabó. Kernel minimum divergence portfolios. 2021. https://arxiv.org/abs/2110.09516.
>
> Richard C. Bradley. Basic properties of strong mixing conditions. A survey and some open questions. Probability Surveys, 2:107–144, 2005. Update of, and a supplement to, the 1986 original.
>
> Wei Biao Wu. Nonlinear system theory: another look at dependence. National Academy of Sciences of the United States of America, 102(40):14150–14154, 2005.

---

> > ### Comment · Reviewer_4v7u · 2025-08-01
> >
> > Thank the authors for their answers. Upon reviewing the authors' responses to all other reviewers, I am satisfied with the provided additional information and experiments. I would like to emphasize the results of additional comparisons with other established methods, such as RuLSIF, Scan-B, and others, as well as more recent methods, including OKCUSUM. Nevertheless, I recommend (if possible) that the authors include detailed discussions about non-Gaussian kernels (from answers to all reviewers) and future work on multiple change point detection, non-i.i.d. case in the main body of the text, and the issue of threshold selection.
> >
> > I have increased my score to 5 (accept).

---

### Official Review · Reviewer_jvfE · 2025-06-27

**Clarity:** 3
**Significance:** 3
**Originality:** 3
**Rating:** 5
**Confidence:** 3

**Summary:**

The paper proposes a scheme for online change point detection using a two-sample test based on maximum mean-discrepancy on a transformed version of the data (feature map based on random Fourier features).  Rigorous guarantees on the performance are obtained including a result showing that it is essentially minimax optimal. A key ingredient of the scheme is performing a logarithmic (in n) number of tests at each time step n.

**Questions:**

- In Section 7, it is mentioned that the theoretical results are valid for any kernel satisfying Assumption 1. Why is the Random Features kernel emphasized, including in the title of the paper — is is due to computational advantages? Could the authors comment on how the performance of the method depends on the choice of kernel (at least empirically)?

- Please clarify how \Lambda is determined — is it just the Lebesgue measure in this case?

- Could you comment on how restrictive Assumption 1 is? It is mentioned that is satisfied by many commonly used kernels, but does it rule out others that are used in practice?

**Ethical Concerns:**

["NO or VERY MINOR ethics concerns only"]

**Final Justification:**

Thanks to the authors for the response. It would be good to include in the paper some of the points clarified in the response, especially  on the choice of the kernel. Overall a solid paper.

**Limitations:**

Yes

**Quality:**

3

**Strengths And Weaknesses:**

The problem studied is interesting and relevant.  There is a substantial literature on onlie change point detection, but the scheme is novel with strong theoretical guarantees, and the numerical results show that the method outperforms the state of the art.  The paper is well-written.

Some potential weaknesses:

- The authors should compare their scheme with the recent framework proposed in https://arxiv.org/abs/2504.09573 which also seems to an approach based on a geometric grid.

- The method appears to be tailored to detect at most one change point. Can it be extended to handle multiple change points?

- The paper emphasizes that other methods require selection of a window parameter. Could the authors explain why this is significant? And Algorithm 1 seems to maintain a window in any case — is that the point that there is no parameter to be optimized?

Typos:

- l. 243: constrains —> constraints
- l. 250: 250.000 —> 250,000
- I think “metrizes” is more standard than “metricizes"

---

> ### Author Rebuttal · Authors · 2025-07-31
>
> We thank the reviewer for the time and effort invested and for the valuable feedback. Below, we answer all questions in detail.
>
> __Restrictiveness of Assumption 1/emphasis on random Fourier features.__ The class of so-called translation-invariant kernel functions, which are kernel functions on $\mathbb R^d$ that only depend on the difference of their inputs (in other words, for a given kernel function $K : \mathbb R^d \times \mathbb R^d \to \mathbb R$ there exists a positive definite function $\psi : \mathbb R^d \to \mathbb R$ such that $K(\mathbf x, \mathbf y) = \psi(\mathbf x - \mathbf y)$; please also refer to Assumption 1 of our manuscript) encompasses many well-known and frequently-used kernel functions, for example, Gaussian kernels, mixtures of Gaussian kernels, inverse multi-quadratic kernels, Matérn kernels, or Laplace kernels. The translation-invariant property permits to approximate the respective kernel function by finite-dimensional feature maps ($\hat z_K$ in the article) through Bochner's theorem, which, in turn, allows the effective online estimation of MMD detailed in the article. The choice of kernel (resp.\ kernel parameter) impacts the online analogue of testing power, that is, a good choice allows to reduce the expected detection delay of our procedure. We emphasize that (i) the results stated in the article hold for all bounded continuous characteristic translation-invariant kernel functions and (ii) the class of such translation-invariant kernel functions is sufficiently large to contain most kernel functions employed on Euclidean spaces in practice. A kernel function on Euclidean space that does not permit a random Fourier feature representation is the unbounded exponential kernel, employed in particular financial contexts \citep{chamakh21kernel}.
>
> We also refer to Section 7 of our manuscript, where we provide references on tuning the kernel w.r.t.\ to testing power (resp.\ a proxy thereof), which in itself is a deep topic outside the scope of this manuscript.
>
>
> __Determining $\Lambda$.__ The existence of the measure $\Lambda$ in Asssumption 1 and (7) follows from Bochner's theorem (see, for example, Wendland 2005, Theorem 6.6), which, besides existence, asserts that $\Lambda$ is a non-negative finite Borel measure (implying that normalization yields a probability measure). For computation, one uses the spectral density associated to $\Lambda$ (in other words, the Radon-Nikodym derivative of $\Lambda$ w.r.t.\ to Lebesgue measure). For many translation-invariant kernel functions these spectral densities are known, see, for example, Sriperumbudur et al. [2010, Table 2], or Rahimi and Recht [2007, Figure 1]. As a concrete example, for the Gaussian kernel, one has a Gaussian spectral density (up to normalization).
>
> We are happy to include the statement of Bochner's theorem in Appendix A.6 ``External statements'' and add a reference in the main part to clarify this point in the manuscript.
>
>
> __Comparison to Moen (2025).__ Unfortunately, we do not think a comparison with the recent work by Moen (2025) would be fair, since their method is designed to detect parametric changes whereas our proposed method is non-parametric. To elaborate: one could easily design a simulation study where Moen's procedure performs poorly, for example, by simulating data where the change occurs in the variance while asking Moen's algorithm to look for a change in the mean. Going in the other direction: if we design a simulation in which Moen's algorithm is run with a contrast function tailored to the specific moment in which the change occurs, then it will have an unfair advantage over the non-parametric methods we currently compare to, which do not have access to this information.
>
> __Detection of multiple change points.__ Detecting multiple change points in a sequential manner is an important problem which has not received much attention in the literature. However, having reviewed the proof of Theorem 3, we observe that the proposed RFF-MMD algorithm can indeed be extended to consistently detect multiple changes in an online manner and we were able to prove a guarantee on the detection delay in this regime, which we are happy to include in the appendices of the manuscript together with a detailed proof.
>
>
> __Benefit of no window parameter.__ As correctly pointed out by the reviewer, one key benefit of having  no window parameter is that there are fewer parameters to tune. A second benefit of having no windows is that our proposed method does not have the minimum detection delay that comes from having to observe at least a window length of data. The third benefit is that we do not "forget" data that falls out of a window. We note that, while we dub the elements of our data structure "windows" in Algorithm 1, these do not correspond to the windowing approach typically employed in online change detection contexts. Still, as each of our "windows" sees a summary of a subset of the data (with all windows capturing the full history), we believe that the naming is appropriate.
>
> We hope these answers fully settle all open questions.
>
> Per August Jarval Moen. A general methodology for fast online changepoint detection. Technical report, 2025. https://arxiv.org/abs/2504.09573.
>
> Bharath K. Sriperumbudur, Arthur Gretton, Kenji Fukumizu, Bernhard Schölkopf, and Gert Lanckriet. Hilbert space embeddings and metrics on probability measures. Journal of Machine Learning Research, 11(50):1517–1561, 2010.
>
> Holger Wendland. Scattered data approximation. Cambridge University Press, 2005.

---

> > ### Comment · Reviewer_jvfE · 2025-08-05
> >
> > Thanks for the response. It would be good to include some of these points in your revised version, especially the clarification on the choice of the kernel. I think this is a solid paper and am happy to retain my score.

---

### Official Review · Reviewer_ArQr · 2025-07-02

**Clarity:** 3
**Significance:** 2
**Originality:** 2
**Rating:** 4
**Confidence:** 3

**Summary:**

The authors consider the problem of online nonparametric change point detection. They consider an algorithm based on a kernel maximum mean discrepancy (MMD) test statistic. Using the Bochner's theorem, they show that (approximate) calculation of the MMD statistic can be reduced to computation of the distance between empirical Fourier features. They suggest a way to update the test statistic in $\mathcal O(r \log n)$ operations on the $n$-th round, which is extremely fast. Here $r$ is the dimension of the RKHS associated with random Fourier features. The authors study theoretical properties of the procedure proving a lower bound on the average running length and a high-probability  upper bound on the detection delay. They also conduct numerical experiments on artificial and real data to illustrate performance of the procedure.

**Questions:**

1. I would appreciate if the authors provided a bit more intuition why Algorithm 1 computes the test statistic from eq. (10) in logarithmic time. Is it true that you save computational time computing averages over segments of length $2^j$ in a recursive manner using averages over segments of length $2^{j - 1}$? If so, can you elaborate on the differences of your procedure compared to online kernel CUSUM, where the authors also update the statistic in a recursive manner?

2. Can you compare your procedure with one of nonparametric change point detection methods based on density ratio estimation (KLIEP, uLSIF, RuLSIF, see S. Liu, M. Yamada, N. Collier, and M. Sugiyama. Changepoint detection in time-series data by relative density ratio estimation. Neural Networks, 43:72–83, 2013)?

3. Can you check performance of your algorithm on other real data sets?

**Ethical Concerns:**

["NO or VERY MINOR ethics concerns only"]

**Final Justification:**

The authors addressed my main concerns during the rebuttal. I decided to keep my score, because the suggested algorithm performs similarly to RuLSIF on the HASC data set. Probably, the method needs further evaluation on real-world data. Nevertheless, the merits of the paper outweight the drawbacks.

**Limitations:**

Yes.

**Paper Formatting Concerns:**

No.

**Quality:**

3

**Strengths And Weaknesses:**

Strengths.

1. While the idea to use random Fourier features was already used by Keriven, Garreau and Poli [22], their NEWMA procedure required a proper choice of the sliding window size. In the present paper, the authors avoid the sliding window technique and obtain a procedure which is almost as fast as NEWMA (up to a $\mathcal O(\log n)$ factor) and uses smaller number of hyperparameters.

2. The authors prove strong theoretical guarantees on performance of the procedure. While study of the test statistic behaviour under the null hypothesis is ubiquitous in the literature on change point detection, not so many papers provide rigorous bounds on the detection delay.


Weaknesses.

1. Numerical experiments on real data are not extensive enough. There is only one real data set.

2. The method is compared with kernel-based approaches only. It is not clear how the method will perform against, say, nonparametric change point detection methods based on density ratio estimation (KLIEP, uLSIF, RuLSIF, see S. Liu, M. Yamada, N. Collier, and M. Sugiyama. Changepoint detection in time-series data by relative density ratio estimation. Neural Networks, 43:72–83, 2013).


Small remark. In Lemma 4, it is better to write ''bounded differences property'', instead of ''self bounding property''. Self-bounding functions are defined differently (see, e.g., Definition 1 in [Boucheron, Lugosi, Massart, Electron. J. Probab., 2009]).

---

> ### Author Rebuttal · Authors · 2025-07-31
>
> We thank the reviewer for the time and effort invested and for the valuable feedback. Below, we answer all questions in detail.
>
> > **Logarithmic complexity for computing (10)**
>
> The computation of (10) requires the computation of $\operatorname{MMD}_{\hat K}$ across all divisions of the dyadic grid, that is, a logarithmic amount. We elaborate the algorithm as follows (in particular, lines 193--195 of the submitted work) and also refer to Appendix A.2 for a schematic representation of the procedure. See below for a comparison to the recursive structure of OKCUSUM.
>
> * **Step 1** Given the data structure employed by Algorithm 1, we may assume that computing MMD at the leftmost division costs $\mathcal O(\log n)$ as one considers the "leftmost" $z$ (given, hence $\mathcal O(1)$) and the sum of all other $z$-s, of which there are at most $\log n$ (the reasoning for the $c$-s is the same), which, for the first computation of MMD results in a time complexity of $\mathcal O(\log n)$ using that (9) is simply their Euclidean distance.
> * **Step 2+** Now, to compute MMD at the next division, we may add the $z$ of the next window to the $z$ of the first and subtract it from the sum of the "right" $z$-s obtained in the previous step. Proceeding equivalently with the corresponding $c$-s, we may again use (9) to obtain MMD, which only gives an additional cost of $\mathcal O(1)$ (as $r$, the number of Fourier features, is fixed).
>
> As we repeat step 2+ at most $\log n$ times, the runtime complexity for obtaining all MMD values is logarithmic in the number of samples. Now, we recall from line 196 that the data structure---which was assumed as given in both steps above---can be maintained in $\mathcal O(\log n)$, which yields the claimed total runtime complexity of $\mathcal O(\log n)$.
>
>
> > **Comparison to recursive structure of OKCUSUM**
>
> The proposed method differs from OKCUSUM in two key aspects. First, OKCUSUM relies on a windowing scheme, essentially instantiating Scan-B statistics for different window sizes, which also requires a pool of reference data known to be from the pre-change distribution (we also refer to our answers to Reviewer "gyF2'' for a quantitative comparison of the respective theoretical results and their implications). Second, OKCUSUM computes the full Gram matrices for estimating MMD. Its recursive update step entails removing the (row,column)-pair corresponding to an old observation and adding the (row,column)-pair corresponding to a new observation. This procedure is illustrated for Scan-B statistics in Appendix A of Li, Xie, Dai, and Song (Sequential Analysis, 2019); OKCUSUM replicates the procedure. Our proposed method does not compute any Gram matrix. Instead, we rely on (9) for approximating MMD using the Euclidean norm of the random Fourier features. While one, in principle, may use recursion for maintaining our dyadic grid structure, we chose to present Algorithm 1 using "for-loops'' instead.
>
>
> > **Comparison to density ratio estimation RuLSIF and other real data sets**
>
> Thank you for the nice reference. While the approaches in Liu, Yamada, Collier, and Sugiyama (Neural Networks, 2013) fundamentally differ from the kernel-based approaches that were considered in our manuscript, we compare them to RuLSIF experimentally on the HASC and MNIST data sets in the following.
>
> * **HASC** The Human Activity Sensing Consortium (HASC) challenge 2011 data set is also considered in the RuLSIF, Scan-B, and OKCUSUM papers. As in the OKCUSUM paper, we consider the change from "walking'' to "staying'' of participant $101$. For pre-processing, we order the corresponding csv-files in the data set lexicographically, omitting the first $1596$ (see below) samples of "walking'', and then concatenating $100$ ""walking'' observations and $100$ "staying'' observations to obtain a total of 10 data sets (with $d=3$) with a single change point each. For obtaining the thresholds of the proposed Online RFF MMD, NewMA, Scan-B statistics, and OKCUSUM, we proceed as elaborated for the MNIST data in Section 6.1, that is, we use a Monte Carlo approach with $\alpha=0.1$ on all $10$ "walking'' data sets. As per the setup in the OKCUSUM paper, for ScanB and OKCUSUM, we set the number of windows $N=14$ and the window length $w=114$, implying that both algorithms receive $14 \times 114 = 1596$ samples from "walking'' upfront. Likewise, we process $1596$ elements with NewMA upfront.
>
> * For RuLSIF (which showed the best performance on HASC in the respective paper), we use the python \texttt{changepoynt} implementation and consider the $l_2$-norm of each three-dimensional observation. We then obtain the change scores for the full data set and select the point with the highest score as predicted change point, as done in Section 4.2 of Liu, Yamada, Collier, and Sugiyama (Neural Networks, 2013). To match their setup, we set the window length to $10$, the number of windows to $50$, and $\alpha=0.1$. With these settings, RuLSIF yields the change points $91, 109,  98,  60,  99, 100,  97,  60,  82,  96$ (up to statistical fluctuations), which we summarize together with the results obtained by the other algorithms in the following table (where we count "too early'' if the change point is flagged before $100$ samples were processed, "miss'' if no change point is flagged, and compute the average over all change points flagged after $100$ observations).
>
> | Algorithm      | Average delay | Too early | Miss |
> |----------------|---------------|-----------|------|
> | Online RFF MMD | 21.86         | 2         | 1    |
> | NewMA          | 34.25         | 1         | 5    |
> | ScanB          | 31.20         | 0         | 0    |
> | OKCUSUM        | 17.44         | 1         | 0    |
> | RuLSIF         | 4.50          | 8         | 0    |
>
> * We emphasize that in the above case all algorithms except for the proposed one and RuLSIF receive $1596$ samples upfront, to use as a reference sample. Even though this fundamentally favors the competitors, our proposed method achieves results that are comparable to those of the other kernel-based approaches. The change points estimated by the RuLSIF algorithm are close to the actual change points in most cases (see the list of change points above), with a tendency of being too early. In practice, one could take this behavior into account by adding a fixed offset. %FK: My guess is that RuLIF estimates the most likely change point posthumously, not really online. I.e., if we have one 1 sample of the post-change distribution, it would not work. We should point this out.
>
> * **MNIST** We recall from Section 6.2 that we consider the changes of $64$ samples of digit $0$ to $64$ samples of digits $1$--$9$. Applying RuLSIF (with the same settings as in the HASC experiments) to the respective $l_2$-norms of the data, the points with the highest change score are $66, 61, 61, 61, 60, 60, 66, 67, 65$; when omitting the $5$ "too early'' cases, the algorithm has an average detection delay of $2.0$.
>
> * **MazurkaBL** We additionally validated our approach on the MazurkaBL data set and refer to our answer to reviewer "4v7u" for details.
>
> We are happy to include these additional results together with all code in the manuscript.
>
>
> > **Performance on other real data sets**
>
> Please see above for additional experiments on the suggested HASC data set.
>
> We hope these answers fully settle all open questions.

---

> > ### Comment · Reviewer_ArQr · 2025-08-02
> > **An additional question about RuLSIF**
> >
> > I would like to thank the authors for clear and constructive feedback. I have an additional question regarding numerical experiments.
> >
> > According to the results on the HASC dataset, RuLSIF has a significantly smaller detection delay, than Online RFF MMD and OKCUSUM. At the same time, it has 8 false alarms while Online RFF MMD and OKCUSUM have only 1-2. For this reason, the presented results do not allow us to compare the performance of the aforementioned algorithms. Can you increase the threshold of RuLSIF in order to reduce the number of false alarms?

---

> ### Author Response · Authors · 2025-08-03
>
> We thank the reviewer for the opportunity to further clarify and extend the experiments concerning RuLSIF.
>
> As stated above, we "obtain the change scores for the full data set and select the point with the highest score as predicted change point, as done in Liu et al. [2013, Section 4.2]". Recall that the RuLSIF change score is based on the density ratio estimated from elements of different windows (called "retrospective subsequence samples" in their article) and is therefore maximized if the pre- and post-change distributions are split accordingly (up to statistical fluctuations; seen in the raw values above, which report at which points the change scores were maximal). Therefore, the moment the algorithm starts processing samples of the post-change distribution the change score starts increasing, before it attains said maximum, and then the score reduces again.
>
> For this reason, decreasing the threshold unfortunately leads RuLSIF to report changes even earlier, increasing the "too early" count in the reported table, while an increase leads to no change being reported.
>
> Like we indicated in the rebuttal above, one can try to manually mitigate these "too early" cases by considering as offset the window length, which ensures that sufficiently many samples of the post-change distribution have been processed.
>
> Hence, in the following, we repeat the above experiment on the HASC data set, varying the window length (30, 40, 50; the latter was already considered above) and the offset (0, w: window length); we keep the other parameters as stated above. As a concrete example, recall from above that, for a window length of 50, the algorithm attained its maximal change scores at $91, 109,  98,  60,  99, 100,  97,  60,  82,  96$. Now, offsetting all values by the window length $50$, we have the "modified" change locations $141, 159, 148, 110, 149, 150, 147, 110, 132, 146$. As the actual change points were at $100$ in all cases, this yields an average delay of $39.2$, $0$ too early cases, and $0$ misses for RuLSIF. We summarize these additional experiments in the following table. Note that a "miss" can not happen due to the change score having at least one maximum.
>
> | Window length | Offset | Average delay | Too early | Miss |
> |---------------|--------|---------------|-----------|------|
> | 50            | 0      | 4.5           | 8         | 0    |
> | 50            | 50     | 39.2          | 0         | 0    |
> | 40            | 0      | 6.0           | 8         | 0    |
> | 40            | 40     | 35.0          | 3         | 0    |
> | 30            | 0      | 11.0          | 9         | 0    |
> | 30            | 30     | 20.375        | 2         | 0    |
>
> These results indicate that even with this additional improvement concerning RuLSIF our proposed method yields competitive results.
>
> We are happy to include a summary of these elaborations in the revised version of our manuscript. Further, we would like to refer to our rebuttal to Reviewer jvfE, Section __Benefit of no window parameter__, where we summarize a few additional benefits that our proposed method features by not relying on a fixed window size.

---

> > ### Comment · Reviewer_ArQr · 2025-08-04
> >
> > Thanks for conducting additional experiments with RuLSIF. I think, the results with Window Length = Offset = 30 are more suitable for comparison with OKCUSUM and Online RFF MMD.

---

### Official Review · Reviewer_gyF2 · 2025-07-05

**Clarity:** 3
**Significance:** 3
**Originality:** 3
**Rating:** 4
**Confidence:** 3

**Summary:**

The paper proposes an online change point detection framework designed to identify abrupt changes in the probability measure of a data stream. The method employs a kernel two-sample test, approximated using random Fourier features. At each time n, the sequence is partitioned into log2​(n) candidate splits using a dyadic grid, and each split is tested using the two-sample test procedure.

**Questions:**

While the limitations of existing online change point detection methods based on fixed-size windows are emphasized, a key drawback of the proposed approach is that its computational complexity increases with sequence length. As a result, suitability to long sequences with no change points is limited. This limitation needs a more detailed discussion.

The description of the algorithm in Section 4.2 and Algorithm 1 is not very easy to follow. Including a schematic diagram in the main paper can improve clarity. In particular, the maintenance steps in step 3 of the algorithm, including window merging operations, are not well explained.

The theoretical results rely on properties of random Fourier features. Theorems 1 and 2 provide bounds on thresholds to achieve a desired average run length or uniform false alarm probability, while Theorem 4 establishes the minimax optimality of the test.

However, the analysis considers the procedure in Equation 10, without including window merging and other management operations. It remains unclear how these components affect performance metrics such as run length, false alarm rate, and detection delay.

Theorem 3 bounds the detection delay, but the expressions are difficult to interpret. How can the thresholds can be selected to achieve a target detection delay when the difference between pre- and post-change distributions is bounded?

The paper lacks a deeper comparative discussion with prior theoretical works. How does the proposed approach compares with existing algorithms such as OKCUSUM, NewMA, or ScanB in terms of run length and detection delay?

The experimental section should explicitly state which thresholds were used.

**Ethical Concerns:**

["NO or VERY MINOR ethics concerns only"]

**Final Justification:**

The paper is clearly written and with solid theoretical results. The additional comparisons to traditional methods such as RulSIF and Scan-B during the discussion was helpful to clarify the experimental performance of the method. The main advantage of the method is the lack of dependence on a window. However, I encourage the authors to provide an in-depth discussion on challenges and limitations of the work in the revised paper, such as the dependence on shift-invariant kernels, multiple change point detection, the non-iid case, and the log(n) complexity increase with the number of samples.

**Limitations:**

The increase of computational complexity with sequence length should be discussed in the limitations. No significant negative societal impact.

**Paper Formatting Concerns:**

No formatting concerns.

**Quality:**

3

**Strengths And Weaknesses:**

Strengths:
- The paper is well written and generally clear.
- Theoretical results seem correct.

Weaknesses:
- The computational complexity of the algorithm increases with the sequence length.
- Certain steps in the algorithm are not described with sufficient clarity.
- It is not evident whether the theoretical results can be used to control the detection delay.

---

> ### Author Rebuttal · Authors · 2025-07-31
>
> We thank the reviewer for the time and effort invested and for the valuable feedback. Below, we answer all questions in detail.
>
> > **Computational complexity increases with the sequence length**
>
> While some authors  (e.g., Chen et al. 2022)  impose the strict requirement that the complexity for processing a new observation may only depend on the number of bits needed to represent the new observation, to the best of our knowledge, there are no algorithms for the problem at hand meeting this requirement, save for those which (implicitly or explicitly) consider a fixed window over which to search for a change---effectively ``forgetting'' data which falls outside of this window. Still, we agree that that the computational complexity of the procedure increasing with the number of data observed can be a limitation.
>
> A key difficulty in achieving $\mathcal{O}(1)$ time complexity for the problem we consider is that, unlike for parametric problems in which under the null (no change) the entire history of the sequence can be represented by a finite-dimensional sufficient statistic, we must keep track of the entire distribution. We also remark that optimal algorithms for online change detection in the parametric change setting generally exhibit $\mathcal{O} (\log n)$ complexity at the $n$-th iteration. See for instance the FOCuS algorithm of Romano et al. [2024] which solves Page's recursion to detect a change in mean in a sequence of i.i.d. Gaussians, running with expected $\mathcal{O} (\log n)$ complexity.
>
> Nevertheless, achieving $\mathcal{O}(1)$ time complexity for the problem at hand is an important open question and an interesting direction for future work.
>
>
> > **Details on merge operations**
>
> We recall from Section 4.2 that each window stores a count $c$ and the sum of the Fourier features $z$ corresponding to the observations that the window captures; the observations themselves are not stored explicitly. The intuition regarding lines 13--22 of Algorithm 1, that is, "the merge'', is as follows: When merging two windows---which, to maintain the dyadic grid structure, happens each time two windows have the same counts (line 16)---it suffices to sum their respective counts (line 17) and to add both their $z$ vectors element-wise (line 18), storing the results in a new window (line 19) and omitting the original two windows (handled by the ``pop'' operations). If no merge happens due to no windows having the same size (line 20), we undo the "pop'' operations of lines 14--15 in line 21.
>
> For a schematic representation of these steps, we refer to Appendix A.2 in the supplement, which we are happy to move to the main part of manuscript, together with the elaborations above.
>
> > **Relationship of (10) and the algorithm's implementation on run length, false alarm rate, and detection delays.**
>
> The window management and merging operations described in Section 4.2 give back exactly the quantity defined in equation (10) with no approximation error. Therefore, the theoretical guarantees obtained by analyzing equation (10) are valid for the outputs of Algorithm 1. We will further emphasize this important point in the final version of the manuscript.
>
>
> > **Application of Theorem 3 to achieve a target detection delay**
>
> Thank you for the question. We appreciate the opportunity to further clarify how the theoretical results in the paper should be understood.
>
> First, we emphasize that the threshold sequence $\lambda_n$ is not chosen to attain a target detection delay, but rather to control the multiple testing analogue of statistical size at a given level.  To elaborate, similar to the methods that we compare against (OKCUSUM, ScanB, NewMA), we work in the sequential testing analogue of the Neyman-Pearson framework. That is, the quantity we seek to control is the multiple testing analogue of statistical size (that is, (i) the average run length defined in line 111, or (ii) the uniform false alarm probability defined in line 113) and subject to either constraint being satisfied, we seek the most powerful procedure (where power is understood in terms of the detection delay). Theorems 1 and 2 show how $\lambda_n$ may be chosen to control either of (i) or (ii) at a desired level, Theorem 3 bounds the detection delay when $\lambda_n$ is chosen to control (ii) according to the result in Theorem 2, and Theorem 4 shows that detection delay guarantee provided in Theorem 3 cannot be improved in general (up to logarithmic terms).
>
> Therefore, Theorem 3 does not provide any guidance on how the thresholds should be chosen beyond the result already given in Theorem 2. However, Theorem 3 details how $r$ (the number of random Fourier features used to approximate the target kernel $K$) should be chosen (recall that the results in Theorems 1 and 2 hold for any choice of $r$). The result is qualitative, and says that when $MMD_K [\mathbb{P}, \mathbb{Q}]$ is bounded away from zero, the number of features should scale as $r = \Theta(\log 1/\alpha)$ with $\alpha$ being the desired uniform false alarm probability in the asymptotic framework where $\alpha \downarrow0$. However, in practice, one may choose $r$ as large as possible subject to computational constraints, keeping in mind that the time and space complexity of RFF-MMD scales like $\mathcal{O} (r \log n)$ at the $n$-th iteration.
>
>
> > **Qualitative comparison to OKCUSUM, NewMA, ScanB w.r.t.\ run length/detection delay**
>
> * **Average run length** All considered algorithms (the proposed Online RFF MMD, OKCUSUM, NewMA, and ScanB) can be calibrated to achieve a desired average run length. The run length analysis for OKCUSUM and ScanB makes use of Siegmund's analysis of boundary crossing probabilities for Gaussian fields whereas our analysis employs simple tail bounds and the law of total probabilities. Nevertheless, we attain the same thresholds up to constant factors.
>
> * **Uniform false alarm probability** However, unlike the proposed method (Theorem 2), neither OKCUSUM, NewMA, nor ScanB offer guidance on calibrating the procedures for controlling the uniform false alarm probability. However, we believe that a peeling argument as in the proof of our Theorem 2 can be used to derive similar guarantees for these methods.
>
> * **Detection delay**  The authors of NewMA and ScanB do not provide a theoretical analysis of the respective detection delay. Moreover, these methods (implicitly and explicitly, respectively) maintain a window, which implies that (i) the detection delay will be at least as long as the window (that is, maintaining a window of size $w$ means the algorithm cannot declare a change earlier than the $w$-th iteration), and (ii) at the $n$-th iteration data with index $n - w$ is effectively "forgotten'' by the algorithm, which can lead to a loss of power. Regarding OKCUSUM, Wei and Xie (Section 4.4) shows that when the OKCUSUM algorithm is run with a maximum window of the order $\Theta ( \log (\gamma) \times  (MMD_K [\mathbb{P}, \mathbb{Q}])^{-2} )$, it will attain an expected detection delay that is asymptotically equivalent to $\log (\gamma) \times  (MMD_K [\mathbb{P}, \mathbb{Q}])^{-2}$ as  $\gamma \uparrow \infty$, where $\gamma$ represents the desired average run length. To emphasize, our analysis is finite sample and is concerned with the setting in which one aims to control the uniform false alarm probability ($\alpha$) under the null, whereas the analysis of Wei and Xie is asymptotic and considers the setting in which one aims to control the average run length. Loosely speaking, in the asymptotic framework where $\alpha \downarrow 0$ and when setting $\gamma = \alpha^{-1}$, our results coincide. Note, however, that Wei and Xie work in the asymptotic setting where a growing number of observations known to be from the pre-change distribution are available to the algorithm, whereas our procedure does not require access to any such information.
>
> > **Thresholds used in experiments**
>
> Regarding the experiments in the main part of the manuscript, we recall from lines 288--290 that we ``compute the thresholds for a given target ARL by processing $10 \times (\text{target ARL})$ samples with each algorithm, repeating for $100$ Monte Carlo (MC) iterations, and computing the $1-1/(\text{target ARL})$ quantile of the resulting test statistics.'' This testing procedure enables a fair comparison of the detection delays because it allows us to ensure that each method is run with the sharpest possible threshold. We are happy to emphasize the latter point in the manuscript. For the additional MNIST experiments provided in Appendix A.3.1, the setup is equivalent to the one described above. The experiments in Appendix A.3.2 use the distribution-free bound of Theorem 1 and therefore require no Monte Carlo iterations.
>
> We hope these answers fully settle all open questions.
>
> Yudong Chen, Tengyao Wang, and Richard J Samworth. High-dimensional, multiscale online changepoint detection. Journal of the Royal Statistical Society. Series B (Statistical Methodology), 84(1):234–266, 2022.
>
> Gaetano Romano, Idris A. Eckley, and Paul Fearnhead. A log-linear non-parametric online change-point detection algorithm based on functional pruning. IEEE Transactions on Signal Processing, 72:594–606, 2024.

---

> > ### Comment · Reviewer_gyF2 · 2025-08-03
> >
> > I thank the authors for the detailed answers. I am satisfied with the clarifications.

---

### Decision · Program_Chairs · 2025-09-17

**Decision:**

Accept (poster)

**Comment:**

The paper proposed a method for online nonparametric change point detection that builds on the kernel Maximum Mean Discrepancy (MMD) test, where the kernel MMD statistic is efficiently approximated using random Fourier features. The work provides a theoretical guarantee of the proposed method. Several weaknesses of the paper, such as the limitation to the kernel type and i.i.d. data, were pointed out by the reviewers. The author's rebuttal addressed the major concerns, and through the discussion, the reviewers reached a consensus on the merit of the work. The authors are encouraged to incorporate the rebuttal into the revised manuscript.